molecular biology/taxonomy and systematics

museum specimens, archival DNA, deep-sea sharks, *Etmopterus pusillus*, *Etmopterus joungi*, taxonomy

**Author for correspondence:**
Nicolas Straube
e-mail: nicolas.straube@uib.no

# Mitochondrial DNA sequencing of a wet-collection syntype demonstrates the importance of type material as genetic resource for lantern shark taxonomy (Chondrichthyes: Etmopteridae)

Nicolas Straube[1], Michaela Preick[2], Gavin J. P. Naylor[3] and Michael Hofreiter[2]

[1]Department of Natural History, University Museum of Bergen, Allégaten 41, 5007 Bergen, Norway
[2]Evolutionary and Adaptive Genomics, University of Potsdam, Karl-Liebknecht-Strasse 24-25, 14476 Potsdam, Germany
[3]Florida Museum of Natural History, University of Florida, Cultural Plaza, 3215 Hull Road, Gainesville, FL 32611-2710, USA

NS, 0000-0001-7047-1084

After initial detection of target archival DNA of a 116-year-old syntype specimen of the smooth lantern shark, *Etmopterus pusillus*, in a single-stranded DNA library, we shotgun-sequenced additional 9 million reads from this same DNA library. Sequencing reads were used for extracting mitochondrial sequence information for analyses of mitochondrial DNA characteristics and reconstruction of the mitochondrial genome. The archival DNA is highly fragmented. A total of 4599 mitochondrial reads were available for the genome reconstruction using an iterative mapping approach. The resulting genome sequence has 12 times coverage and a length of 16 741 bp. All 37 vertebrate mitochondrial loci plus the control region were identified and annotated. The mitochondrial NADH2 gene was subsequently used to place the syntype haplotype in a network comprising multiple *E. pusillus* samples from various distant localities as well as sequences from a morphological similar species, the shortfin smooth lantern shark *Etmopterus joungi*. Results confirm the almost global distribution of *E. pusillus* and suggest *E. joungi* to be a junior synonym of *E. pusillus*. As mitochondrial DNA often

represents the only available reference information in non-model organisms, this study illustrates the importance of mitochondrial DNA from an aged, wet collection type specimen for taxonomy.

## 1. Introduction

Type material deposited in museum collections is generally essential for taxonomic purposes. Fixation and preservation, especially in wet collections, damages DNA to a degree that these samples are rarely available for standard DNA sequencing analysis. However, recent methodological advances show that archival DNA from museum wet collection specimens is widely accessible by the application of ancient DNA methods (e.g. [1–4]).

In scientific ichthyological collections, specimen preservation in 75% ethanol commonly causes DNA sequence damage. Our recent increase in taxonomic knowledge partially fuelled by DNA barcoding [5], for example, often results in pending taxonomic questions, where type material DNA sequence information is crucial to answer these. Especially mitochondrial DNA is of major interest due to growing reference databases as a source for species delimitation based on DNA sequence differences.

Deep-sea shark taxonomy is a vivid example of challenges arising with increased sampling activities where new sequence information is generated producing new taxonomic questions. One group of focus are the lantern sharks (Etmopteridae), where several new species were described recently (e.g. [6–9]) and new DNA sequence information is also available (e.g. [10]).

Etmopteridae are the most species-rich shark family with 54 species described in four genera [11]. The species-rich genus *Etmopterus*, containing 45 species [11], is subdivided into four monophyletic subgroups: the *E. spinax*, *E. gracilispinis*, *E. pusillus* and *E. lucifer* clades [12].

Despite their worldwide distribution and species diversity, their phylogenetic relationships have not been studied extensively at the species level. Exact distribution ranges are often poorly known due to the scarcity of specimens available, or a lack of comparative studies that incorporate samples from large geographical areas. While some species appear localized, such as *E. lailae* Ebert, Papastamatiou, Kajiura & Wetherbee, 2017 or *E. marshae* Ebert & Van Hees, 2018, others are widespread, for example, *E. granulosus* (Günther 1880) or *E. viator* Straube, 2011 [13–15]. Given morphological similarities between closely related species and a lack of documentation of morphological changes throughout ontogeny and/or differences between the sexes, the geographical origin of a sample is often relied upon to distinguish among congeners that are morphologically similar.

The smooth lantern shark, *Etmopterus pusillus* (Lowe, 1839), typifies challenges outlined above. As its common name indicates, it is characterized by smooth skin, a feature caused by block-like rather than the typical tooth-like dermal denticles, a rare feature among etmopterids in general and within the *E. pusillus* clade only shared by *E. joungi* Knuckey *et al*. 2011 [16] and the more distantly related blurred smooth lantern shark, *Etmopterus bigelowi* Shirai & Tachikawa 1993 [10,12,17]. The smooth lantern shark was originally described from Madeira in the Northeast Atlantic Ocean. After its description in 1839, morphologically similar specimens were reported from numerous geographically distant locations far from the type locality, suggesting the species was widespread. Shirai & Tachikawa [18] were the first to explore the taxonomic status of the morphologically similar specimens. These authors described a new species, *E. bigelowi* and commented on the distribution range of *E. pusillus* as occurring almost globally by declaring the Northwest Pacific species *E. frontimaculatus* Pietschmann 1907 as a junior synonym to *E. pusillus*. In 2011, however, Knuckey *et al*. [16] described a morphologically similar species from the Northwest Pacific (Taiwan), the shortfin smooth lantern shark *E. joungi*. Later studies based on the analyses of mitochondrial NADH dehydrogenase subunit 2 (NADH2) gene sequences showed that mitochondrial haplotypes of *E. pusillus* from both the Pacific and Atlantic Ocean mix along with specimens identified as *E. joungi* from Taiwan [10]. The authors suggested that the species status of *E. joungi* be reviewed to clarify whether *E. joungi* represented a subpopulation of *E. pusillus* or indeed a distinct species. Here, we analyse data using an ancient DNA approach to sequence archival mitochondrial DNA of one of the syntypes of *E. pusillus* collected in 1903 in the Northwest Pacific off Japan, which represents a specimen of the former type series of *E. frontimaculatus*.

In this study, we reconstruct the full mitochondrial genome of the 116-year-old ethanol-preserved type specimen with an unknown fixation history from shotgun short-read sequences. Further, we use sequence information from the mitochondrial NADH2 gene for placing the syntype haplotype in a phylogenetic network of a broader sampling of *E. pusillus* and *E. joungi* NADH2 sequences. We use

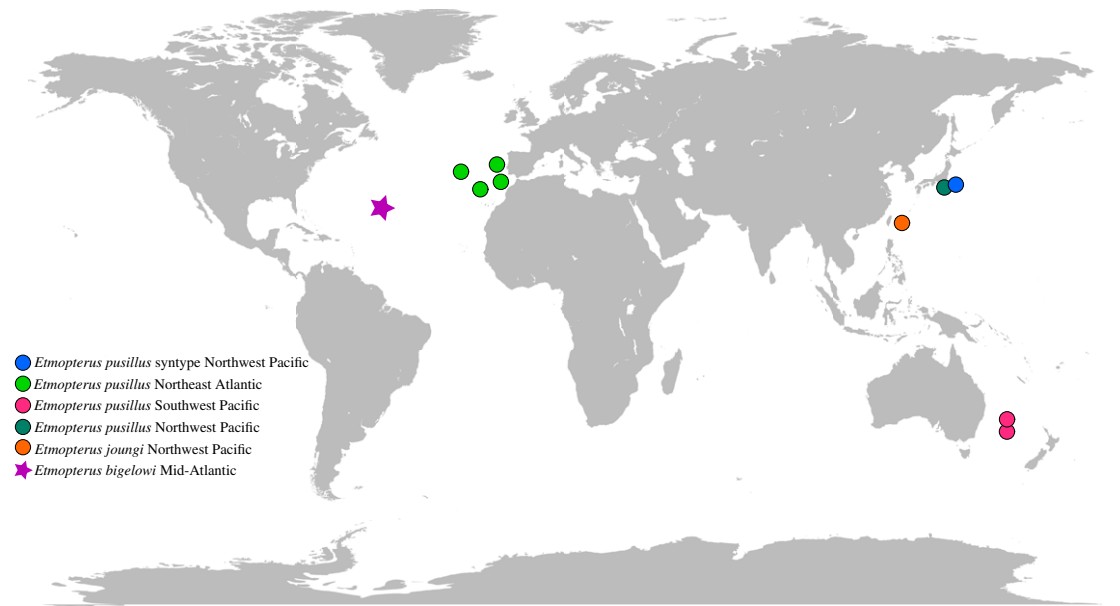

**Figure 1.** Overview of sampling locations.

the sequence information to estimate the haplotype diversity from both species and discuss the distribution range of *E. pusillus*.

# 2. Material and methods

## 2.1. Mitochondrial genome reconstruction of the *E. pusillus* syntype museum specimen

Tissue from the syntype was sampled during a research visit by N.S. to the Museum of Natural History (MNH) in Vienna, Austria, in 2017. The syntype series comprises eight specimens catalogued with three collection lot numbers, i.e. NMW-78526, NMW-65832 and NMW-61473. The tissue sample analysed herein was dissected from specimen NMW-78526-15. Muscle tissue (67.7 mg) was removed by MNH staff with a sterile scalpel and tweezers and kept in the original preservation fluid until DNA extraction. The specimen was originally collected in 1903 off Yokohama (Japan) in the Northwest Pacific (figure 1). By the time of analysis in 2019, it was 116 years old. Details on its fixation and preservation history are unknown, but we assume that it was preserved as a wet collection specimen in supposedly diluted ethanol since it arrived at the museum.

Due to the age and preservation of the syntype specimen, DNA was extracted, and a single-stranded library constructed for this specimen, described in detail in Straube, Lyra *et al.* [4]. In brief, Straube, Lyra *et al.* [4] applied the Guanidine treatment DNA extraction approach modified from [19–21] using the 67.7 mg of muscle tissue. Thereafter, DNA fragments were converted into a single-stranded DNA library following the protocol in Gansauge & Meyer [22]. Library fragment size was checked using an Agilent Tapestation®. DNA extraction and library preparation were conducted in a dedicated historical DNA laboratory at the University of Potsdam following the recommendations in Fulton & Shapiro [23]. The library was test-sequenced on an Illumina NextSeq® instrument with 1 758 548 75 bp single-end reads using the 500/550 High Output v. 2.5 (75 cycles, Illumina 20024906) kit and custom sequencing primers [24]. Reads were quality filtered and trimmed using cutadapt [25]. The presence of target DNA was checked with Fastqscreen [26] using a set of reference genomes to evaluate portions of both contaminant and target DNA. The *Etmopterus spinax* (Linnaeus, 1758) transcriptome derived from Delroisse *et al.* [27] and the *E. pusillus* mitochondrial genome from Chen *et al.* [28] were used as custom target references for checking for target DNA in the syntype DNA library (see Straube, Lyra *et al.* [4] for further details).

Using these data, we started an initial experiment with Mitobim [29], where we first assembled reads using MIRA [30] and then iteratively mapped reads. The *E. pusillus* mitochondrial genome (KU892588; [28]) was used as a seed for mapping in the first iteration. Subsequently, we counted the number of reconstructed sites in the last iteration. For an approximate estimate of mitochondrial nucleotides present in the test sequencing data, we divided the number of recovered mitochondrial nucleotides by

the total number of raw reads from test sequencing (1 758 548 reads) resulting in the theoretical number of mitochondrial nucleotides per read. Dividing the length of the mitochondrial genome of *E. pusillus* (16 729 bp; [28]) by the mitochondrial nucleotides per read allowed us to estimate the number of further reads necessary to be able to reconstruct the full mitochondrial genome sequence, guiding subsequent deeper sequencing of the DNA library.

Reads with a minimum length of 25 bp were trimmed allowing for an overlap of 1 bp between read and adapter. To reconstruct the full mitochondrial genome sequence from the trimmed reads, we proceeded as described above for the test sequencing reads: first we assembled the reads de novo using MIRA. Thereafter, we used the iterative mapping approach in Mitobim. We allowed Mitobim a maximum of 10 iterations and lowered the k-mer size from the default of 31 to 21 after experiments to optimize the k-mer size. Geneious R7 (https://www.geneious.com) was used to compute the coverage by importing the .caf files resulting from the Mitobim analysis and annotate the mitochondrial genome based on the available annotation from KU892588 [28]. To align the reconstructed mitochondrial genome sequence from the syntype to the reference Genbank mitochondrial sequence, we used the Geneious consensus aligner. We estimated archival mitochondrial DNA damage patterns from the mitochondrial readpool used for reconstructing the mitochondrial genome in the last iteration of Mitobim. First, BWA [31] was used to map these reads to the *Etmopterus pusillus* reference KU892588. Samtools [32] was subsequently used to remove low mapping quality reads and PCR duplicates. The resulting file was input to Mapdamage v. 2.0.7 [33], which was run under default settings to estimate the damage patterns of the archival mitochondrial syntype DNA. The same set of reads was further used to map reads back to the reconstructed mitochondrial genome of the syntype using BWA to test if reads can be successfully and unambiguously mapped. BWA was run as described for the total sum of reads. Samstat v. 1.5.1 [34] was used to compare the percentage of mapped reads used in Mitobim and mapping all reads to the reconstructed mitochondrial genome sequence of the syntype.

MITOS ([35]; http://mitos.bioinf.uni-leipzig.de/index.py) was used to gain further annotation information such as plus and minus strand loci coordinates. Further, it was used to check the arrangement of genes on the mitochondrial genome.

## 2.2. NADH2 sequencing of *E. pusillus* and *E. joungi* fresh material

In the course of the Chondrichthyan Tree of Life project (Ctol, https://sharksrays.org/), numerous samples of *Etmopterus* were collected on diverse field trips or contributed by museum tissue collections for sequencing the NADH2 gene in an approach to create a mitochondrial reference library at the species level. Sampling locations are shown in figure 1. At least one specimen of each putative species was identified morphologically by contributors of Ctol and deposited in a scientific collection [17]. Haplotypes of these vouchered specimens were used as taxonomic references for further species assignations of NADH2 haplotypes of samples contributed to Ctol. DNA was extracted from fresh tissue samples using the E.Z.N.A. Tissue DNA Kit (Omega Biotek, Inc., Norcross, GA, USA). Extracted genomic DNA was used to amplify the NADH dehydrogenase subunit 2 gene of the mitochondrial DNA via PCR. A single set of universal primers [36] designed to bind to the ASN and ILE tRNA regions of the mitochondrial genome was used to amplify the target fragment. PCR reactions were carried out in 25 µl volume comprising 0.3 µM primers, 2.5 mM MgCl$_2$, 200 µM dNTP each, 1× Ex Taq buffer, 0.25 U Takara Ex Taq (Takara, Mountain View, CA, USA) and 50–100 ng template DNA. The reaction mixture was denatured at 94°C for 3 min. Thereafter, 35 PCR cycles were carried out comprising denaturation at 94°C for 30 s, annealing at 48°C for 30 s and extension at 72°C for 90 s. PCR products were purified with ExoSAP-IT (USB, Cleveland, OH, USA), and bi-directionally Sanger sequenced using BigDye® Terminator chemistry on an ABI 3730xl genetic analyser (Applied Biosystems®, Life Technologies, Grand Isand, USA), at Retrogen Inc. Custom DNA Sequencing Facility (San Diego, CA, USA). DNA sequences were edited using Geneious® Pro v. 6.1.7 (Biomatters Ltd Auckland, New Zealand). The edited sequences were translated to amino acids and aligned with corresponding NADH2 sequences from representatives of closely related species using the MAFFT [37,38] module within Geneious. The aligned amino acid sequences were translated back in frame to their original nucleotide sequences, to yield a nucleotide alignment. The full protein-coding alignment was 1044 bp in length.

## 2.3. NADH2 haplotype network

To place the syntype haplotype in a larger framework of *E. pusillus* and *E. joungi* sequences, we aligned the full mitochondrial genome syntype sequence to an alignment of mitochondrial NADH2 sequences derived from the Chondrichthyan Tree of Life project comprising 36 specimens identified as *E. pusillus* and *E. joungi*

from different localities as well as a single NADH2 sequence of the sister species of both taxa, *Etmopterus bigelowi* (table 1). Thereafter, the NADH2 sequence was extracted from the mitochondrial genome syntype sequence and re-aligned to the NADH2 sequence alignment. As described in §2.2, the full alignment was translated to amino acids to check for stop codons and backtranslated in frame.

PopART v. 1.7 [39] was used to reconstruct a haplotype network from the NADH2 sequences. PopART was run using the Median Joining inference under default settings using a provided trait file coding for locality information of samples.

# 3. Results

## 3.1. Syntype archival DNA

Using the test sequencing data from Straube, Lyra *et al.* [4] for the initial Mitobim analysis, 11 122 mitochondrial nucleotides were recovered after 13 iterations. Assuming a total mitochondrial genome sequence length of 16 729 bases (equalling the length of the reference mitochondrial genome KU892588 [28]), we estimated that a maximum of 13 940 833 reads would need to be sequenced for a minimum fivefold coverage of the mitochondrial genome given the preliminary data. To comply with this target, 9 901 517 additional reads were subsequently sequenced. After trimming, 8 188 491 reads were available for further analysis. The distribution of sequencing reads shows a high level of fragmentation with the majority of reads below a length of 50 bp (figure 2), while the average insert size of the library is 30.5 nucleotides. In combination with the trimmed test sequencing reads from Straube, Lyra *et al.* [4], a total of 8 970 773 trimmed reads for the reconstruction of the mitochondrial genome and further analysis were available.

## 3.2. Syntype mitochondrial DNA and genome reconstruction

After assembling trimmed reads with Mitobim, the full mitochondrial genome was recovered from a total of 4559 mitochondrial reads after three iterations. The analysis resulted in a single consensus sequence of 16 741 bp with an average per base coverage of 12.12 exceeding our estimates from the preliminary test sequencing data (table 2 and figure 3). Mapping reads used for mitochondrial genome reconstruction by Mitobim back to the reconstructed syntype mitochondrial genome showed that 3808 (81.8%) of 4654 reads mapped; 80.5% of reads have a mapping quality of greater than or equal to 30. Mapping all available 8 188 491 trimmed reads derived from shotgun sequencing onto the reconstructed mitochondrial genome resulted in a similar number (3850) of reads mapping to the reconstructed genome sequence.

Estimated damage patterns of the archival mitochondrial DNA show an elevated rate of cytosine to thymine substitutions at the 5′ end of reads indicating cytosine deamination. The level of deamination is below 5% (figure 4). Annotation of the mitochondrial genome with Geneious allowed all coding genes, 12S rRNA, 16S rRNA and tRNAs to be identified, as well as defining the control region. Table 2 shows the summary statistics for the mitochondrial genome sequence. Based on the reference annotation from Chen *et al.* [28], we confirm that the syntype mitochondrial genome comprises the complete set of 37 vertebrate mitochondrial genes, i.e. 13 protein-coding genes, two ribosomal RNAs and 22 transfer RNAs as well as the control region. Most loci were encoded on the H strand except for eight tRNAs (trnQ, trnA, trnN, trnC, trnY, trnS2, trnE, trnP) and a single protein-coding gene, NAD6. The sequence of loci is shown in figure 5.

## 3.3. Haplotype network

After manual editing of the NADH2 alignment in Geneious, 1044 sites each of 38 samples in total were available for the analysis (table 1). PopART detected 27 haplotypes with a nucleotide diversity pi of 0.004. There are 100 segregating sites and 22 parsimony-informative characters. The network visualized in figure 6 shows two closely related haplotype groups, one comprising haplotypes from the Northeast Atlantic as well as the Southwest Pacific, while the other one includes haplotypes from the North and Southwest Pacific including the syntype of *E. pusillus* and *E. joungi* specimens. Both groups are separated by two mutational steps. The syntype haplotype is identical to an *E. joungi* haplotype as well as a Northwest Pacific *E. pusillus* haplotype. The sister species, *E. bigelowi*, is separated from the *E. pusillus* and *E. joungi* cluster by 71 mutational steps (figure 6).

**Table 1.** Overview of samples used for the NADH2 haplotype network reconstruction.

| genus | species | locality | tissue sample number | accession (Genbank/Dryad) | source |
|---|---|---|---|---|---|
| Etmopterus | joungi | Northwest Pacific, Taiwan (Ta-Shi fish market) | GN10144 | KF927858 | Straube et al. [10] |
| Etmopterus | joungi | Northwest Pacific, Taiwan (Ta-Shi fish market) | GN10145 | KF927859 | Straube et al. [10] |
| Etmopterus | joungi | Northwest Pacific, Taiwan (Ta-Shi fish market) | GN10140 | KF927857 | Straube et al. [10] |
| Etmopterus | joungi | Northwest Pacific, Taiwan (Ta-Shi fish market) | GN10148 | MZ395856 | this study |
| Etmopterus | pusillus | Northeast Atlantic, Azores, Portugal | GN6543 | MZ395857 | Naylor et al. [17] |
| Etmopterus | pusillus | Northeast Atlantic, Azores, Portugal | GN6548 | MZ395858 | Naylor et al. [17] |
| Etmopterus | pusillus | Northeast Atlantic, Azores, Portugal | GN6550 | MZ395859 | Naylor et al. [17] |
| Etmopterus | pusillus | Northeast Atlantic, Azores, Portugal | GN6552 | JQ518964 | Naylor et al. [17] |
| Etmopterus | pusillus | Northeast Atlantic, Azores, Portugal | GN6555 | MZ395860 | Naylor et al. [17] |
| Etmopterus | pusillus | Northeast Atlantic, Azores, Portugal | GN6563 | MZ395861 | Naylor et al. [17] |
| Etmopterus | pusillus | Northeast Atlantic, Azores, Portugal | GN6578 | MZ395862 | Naylor et al. [17] |
| Etmopterus | pusillus | Northeast Atlantic, Portugal | GN6603 | MZ395863 | Naylor et al. [17] |
| Etmopterus | pusillus | Northeast Atlantic, Portugal | GN6620 | MZ395864 | Naylor et al. [17] |
| Etmopterus | pusillus | Northeast Atlantic, Morocco | GN12128 | MZ395877 | this study |
| Etmopterus | pusillus | Northeast Atlantic, Portugal | GN3772 | MZ395871 | Naylor et al. [17] |
| Etmopterus | pusillus | Northeast Atlantic, Portugal | GN3771 | KF861692 | Straube et al. [15] |
| Etmopterus | pusillus | Northeast Atlantic, Portugal | GN3770 | MZ395872 | Naylor et al. [17] |
| Etmopterus | pusillus | Northeast Atlantic, Portugal | GN3769 | MZ395873 | Naylor et al. [17] |
| Etmopterus | pusillus | Northeast Atlantic, Portugal | GN3767 | MZ395874 | Naylor et al. [17] |
| Etmopterus | pusillus | Northeast Atlantic, Portugal | GN3766 | KF861691 | Straube et al. [15] |
| Etmopterus | pusillus | Northeast Atlantic, Portugal | GN3765 | MZ395875 | Naylor et al. [17] |
| Etmopterus | pusillus | Northeast Atlantic, Portugal | GN6624 | MZ395878 | Naylor et al. [17] |
| Etmopterus | pusillus | Southwest Pacific, New South Wales, Australia | GN11335 | MZ395866 | this study |

(Continued.)

**Table 1.** (*Continued.*)

| genus | species | locality | tissue sample number | accession (Genbank/Dryad) | source |
|---|---|---|---|---|---|
| *Etmopterus* | *pusillus* | Southwest Pacific, New South Wales, Australia | GN11334 | MZ395867 | this study |
| *Etmopterus* | *pusillus* | Southwest Pacific, New South Wales, Australia | GN11331 | MZ395868 | this study |
| *Etmopterus* | *pusillus* | Southwest Pacific, New South Wales, Australia | GN11330 | MZ395869 | this study |
| *Etmopterus* | *pusillus* | Southwest Pacific, New South Wales, Australia | GN11329 | MZ395870 | this study |
| *Etmopterus* | *pusillus* | Southwest Pacific, New South Wales, Australia | GN2614 | MZ395876 | Naylor *et al.* [17] |
| *Etmopterus* | *pusillus* | Southwest Pacific, New South Wales, Australia | GN2613 | MZ395879 | Naylor *et al.* [17] |
| *Etmopterus* | *pusillus* | Southwest Pacific, New South Wales, Australia | GN4951 | MZ395880 | Naylor *et al.* [17] |
| *Etmopterus* | *pusillus* | Southwest Pacific, New South Wales, Australia | GN11328 | MZ395881 | this study |
| *Etmopterus* | *pusillus* | Southwest Pacific, New South Wales, Australia | GN7396 | MZ395884 | this study |
| *Etmopterus* | *pusillus* | Northeast Pacific, Suruga Bay, Japan | GN7426 | MZ395865 | this study |
| *Etmopterus* | *pusillus* | Northeast Pacific, Suruga Bay, Japan | GN7436 | MZ395882 | this study |
| *Etmopterus* | *pusillus* | Northeast Pacific, Suruga Bay, Japan | GN7435 | MZ395883 | this study |
| *Etmopterus* | *pusillus* | Northeast Pacific, Yokohama, Japan (Syntype) | Epusisyn | https://doi.org/10.5061/dryad.kd51c5b59 | this study |
| *Etmopterus* | *pusillus* | East China Sea | na | KU892588 | this study |
| *Etmopterus* | *bigelowi* | Mid-Atlantic | GN5026 | MZ395885 | Naylor *et al.* [17] |

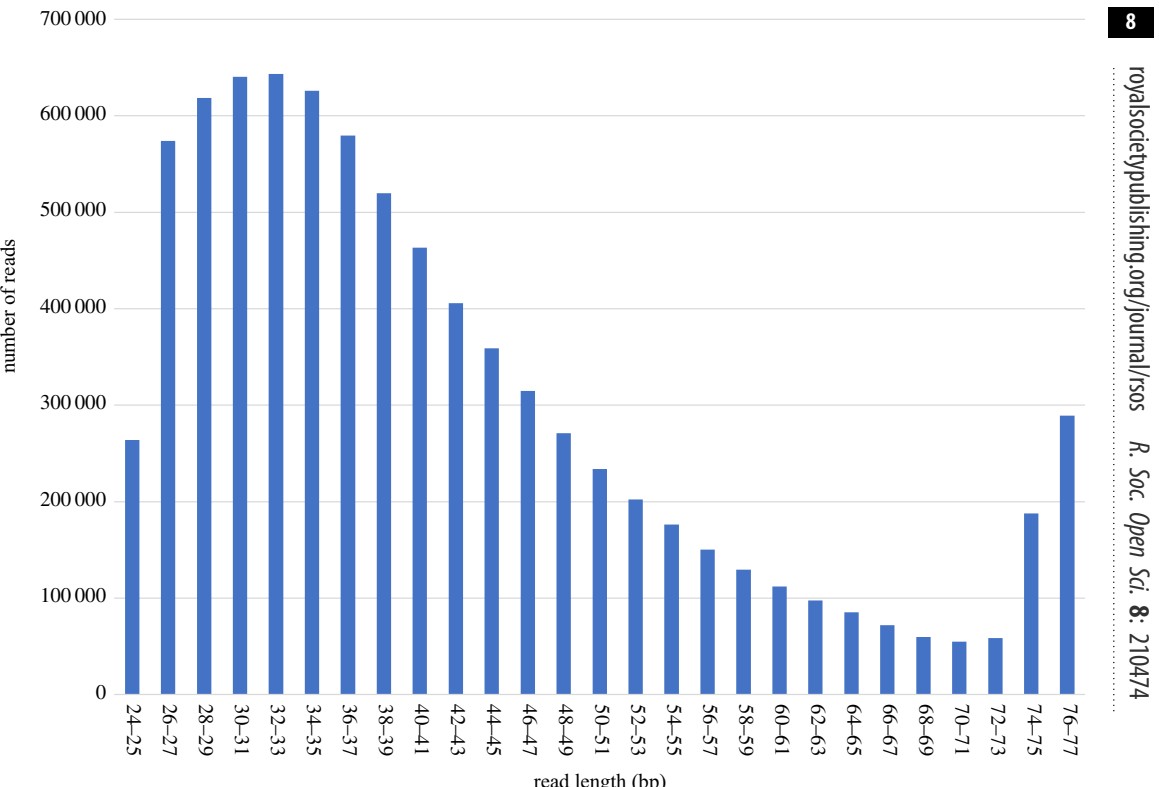

**Figure 2.** Read length distribution of trimmed reads ($N = 8\,188\,491$) derived from the *E. pusillus* syntype specimen. The majority of reads are below 50 bp in length.

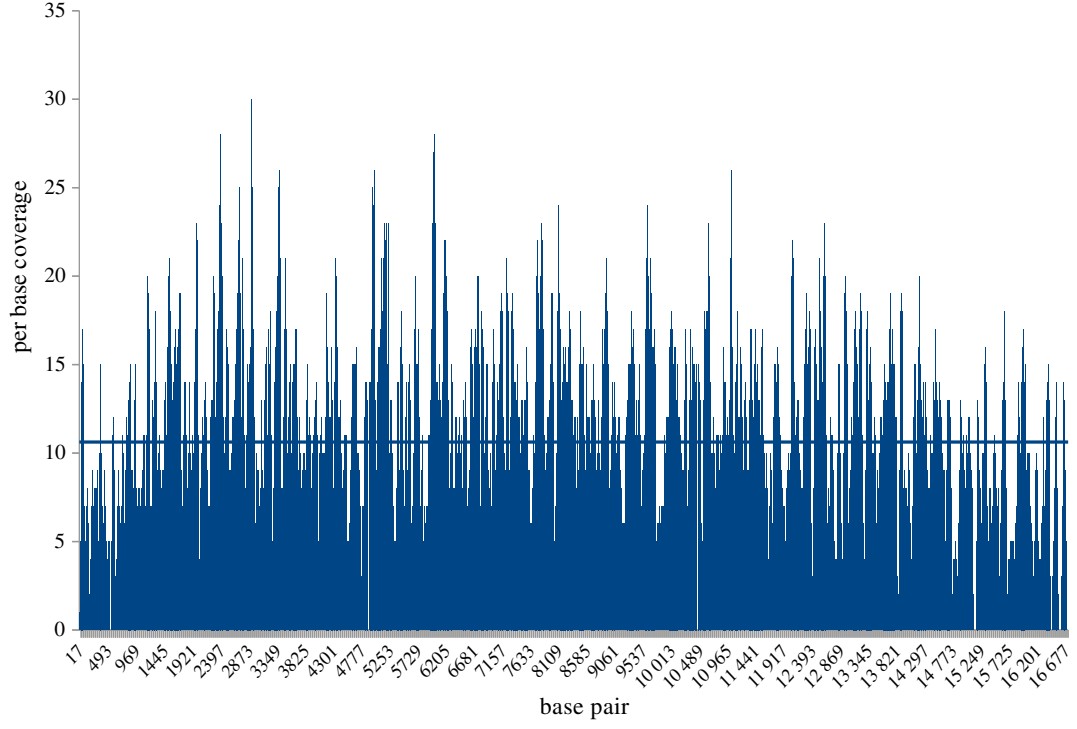

**Figure 3.** Per base coverage plot of the *E. pusillus* syntype NMW-78526-15 mitochondrial genome. Horizontal line shows the median.

**Table 2.** Summary statistics for the reconstructed mitochondrial genome sequence of the *E. pusillus* syntype NMW-78526-15.

| length (bp) | GC content (%) | A (%) | C (%) | G (%) | T (%) | N (%) | average per base coverage |
|---|---|---|---|---|---|---|---|
| 16 741 | 37.2 | 31.1 | 22.8 | 14.3 | 31.5 | 0.4 | 12.12 |

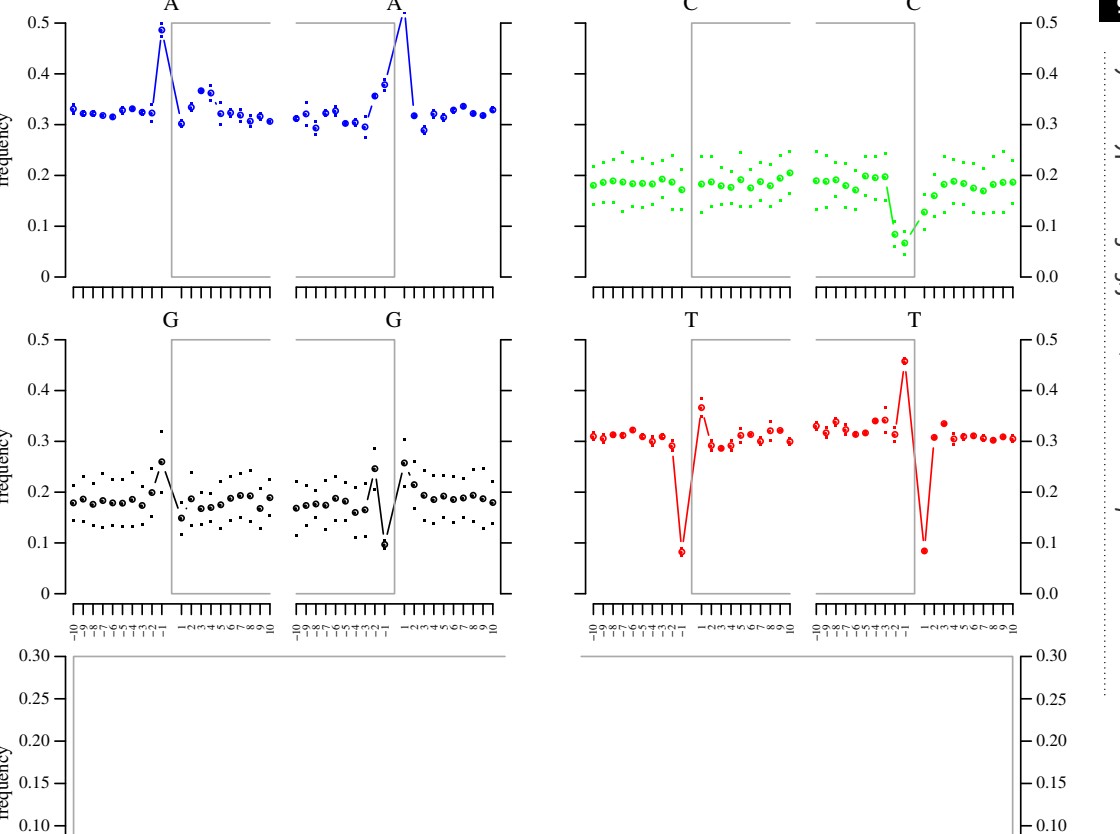

**Figure 4.** DNA damage patterns of the 116-year-old *E. pusillus* syntype archival mitochondrial DNA. Sequencing reads in bottom panel show increased C to T substitutions (red) at the 5′ ends. Blue: G to A substitutions. Grey: all other substitutions. Upper four plots (blue, green, black and red) show base frequencies of inside and outside of reads (inside reads indicated by brackets).

# 4. Discussion

## 4.1. Syntype mitochondrial DNA recovery

As previously described for this and other wet collection samples [4], the fragmentation level of the sequenced archival DNA is high, with an average fragment length of 42.68 bp indicating substantial degradation of the DNA (figure 2). The exact causes of this drastic and relatively rapid degradation remain unknown; however, the storage conditions in an ethanol solution probably fuelled fragmentation through hydrolysis.

The ancient DNA approach applied to the sample in Straube, Lyra *et al*. [4] was successful as target DNA was detected. Our initial iterative mapping experiment with Mitobim showed that enough mitochondrial reads should be recovered with approximately 10 million further shotgun-sequencing reads which proved successful. The damage estimates of the recovered mitochondrial DNA are comparable to findings in other wet collections samples [4], i.e. deamination is present but to a limited degree (figure 4). Recovering the full mitochondrial genome from shotgun sequencing saves time-consuming and costly target capture approaches, which demand additional laboratory working time besides prior bait design and production. Test sequencing of libraries using a limited number of reads as a first step to check for target DNA and contamination levels has also been successfully applied as an adequate strategy for sequencing dry museum and herbarium specimens (e.g. [40]). Our subsequent approach allowed for generating key reference data for answering pending taxonomic questions.

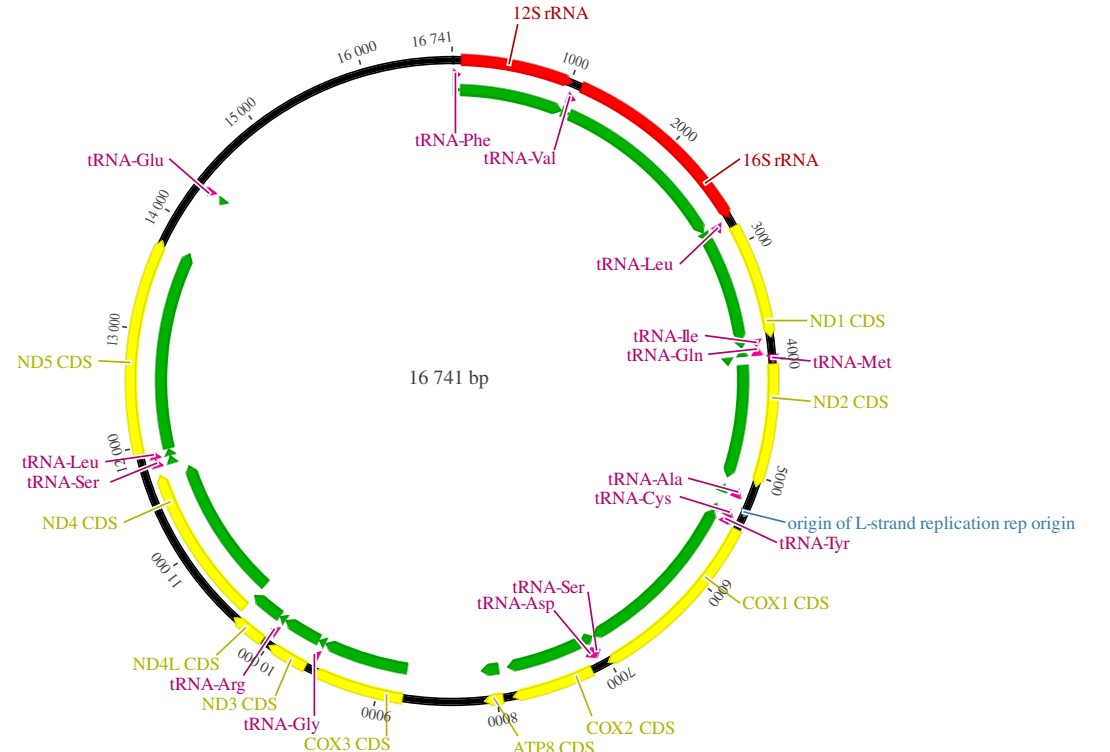

**Figure 5.** Annotated circular mitochondrial genome sequence of the *E. pusillus* syntype NMW-78526-15. Annotation reference: GenBank accession KU892588 [28].

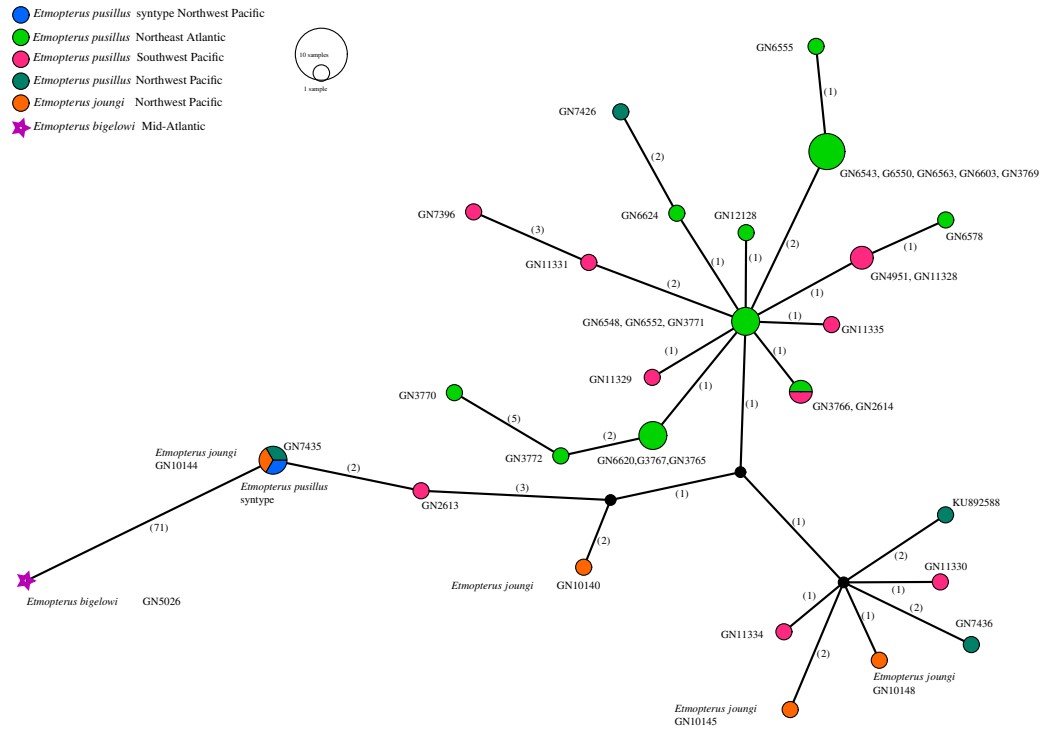

**Figure 6.** Haplotype network of NADH2 sequences derived from *E. pusillus*, *E. joungi*, the *E. pusillus* syntype specimen as well as *E. bigelowi*. Numbers in parentheses indicate mutational steps.

## 4.2. Distribution of *Etmopterus pusillus* and taxonomic implications

Within Etmopteridae, large distribution ranges are known for several species [13–15]. Our results support the notion by Shirai & Tachikawa [18] that *E. pusillus* is a widespread species. In fact, it is

distributed almost circumglobally. The lack of molecular differences across vast geographical distances is striking (figure 6). Indeed, Southwest Pacific mitochondrial haplotypes cluster together with both Northeast Atlantic (close to the type locality of *E. pusillus*) and Northwest Pacific haplotypes. Substantial long-distance dispersal potential has been suggested as an explanation to account for the lack of population structure in several deep-sea shark species [41–44], including *Etmopterus* species [14].

Mitochondrial sequence divergence is often used for species delimitation in sharks (e.g. [10,17]). Admixture of mitochondrial haplotypes is, therefore, impacting taxonomy. North Pacific mitochondrial haplotypes of the Pacific sleeper shark *Somniosus pacificus* Bigelow & Schröder, 1944 are also found in specimens assigned to the Antarctic sleeper shark *Somniosus antarcticus* Whitley, 1939, for example, which questions the taxonomic status of *S. antarcticus* [41,45,46]. The results presented herein also impact the taxonomy of *E. joungi*. Our haplotype network shows that the *E. pusillus* syntype shares a haplotype with a specimen identified previously as *E. joungi* in [10] making a distinction of the two species with mitochondrial NADH2 sequence data impossible. This mitochondrial gene sequence, however, was demonstrated to be highly diagnostic for species-level distinction in chondrichthyan fishes including *Etmopterus* [10,17]. The expectation of monophyly is not met for *E. joungi*. In their description of *E. joungi*, Knuckey *et al.* [16] report on intraspecific geographical morphological variation within their comparative material of *E. pusillus* analysed. The detected morphological differences between *E. pusillus* and *E. joungi* in Knuckey *et al.* [16] may represent further intraspecific variation. We, therefore, suggest that *E. joungi* is a junior synonym of *E. pusillus*. Our DNA sequence-based results support the synonymy of *E. frontimaculatus* with *E. pusillus*, first suggested by Shirai & Tachikawa [18] based on morphological evidence.

Even though etmopterids are not directly targeted in fisheries, the shift of commercial fisheries to fishing grounds in deeper waters [47,48] poses a threat to slow-growing and late-maturing species with few offspring such as *E. pusillus*. Detailed information on the distribution ranges of such species is essential for precise taxonomic assessment and is the basis for correct evaluation for protection and management strategies. About *E. pusillus*, our results of an almost global marine distribution give hope for cautious optimism regarding its future. Since little is known about geographical ranges and population sizes of many shark species, especially deep-sea ones, filling these gaps in our knowledge is of prime importance, as only then, conservation efforts can be focused on those species that are under the greatest danger of extinction.

Data accessibility. NADH2 sequence data analysed in this study are available in Genbank; see table 1 for accession numbers. Sequence data of the syntype are archived from the Dryad Digital Repository: https://doi.org/10.5061/dryad.kd51c5b59 [49].

Authors' contributions. N.S. and M.H. designed the study and analysed the data. N.S. and M.P. conducted laboratory work. G.J.P.N. provided NADH2 sequences for the network analysis. N.S. wrote and drafted the manuscript with contributions from all authors. The final version of the manuscript was approved by all authors.

Competing interests. We declare we have no competing interests.

Funding. This work was funded by the German Research Foundation (DFG; project number 351649567 to N.S. and M.H. within the DFG SPP 1991 'Taxon-Omics'). This project was further supported by the National Science Foundation (NSF), grant DEB-01132229 to G.J.P.N.

Acknowledgements. We would like to express our sincere thanks to Ernst Mikschi and Anja Palandacic at the Museum of Natural History in Vienna for the sampling opportunity of the *E. pusillus* syntype. Further, Axel Barlow, Stefanie Hartmann, Marianna Lyra, Isabella Stöger and Johanna Paijmans are acknowledged for advice regarding data analysis. Dave Ebert is thanked for discussion of taxonomic results. Three anonymous reviewers are thanked for constructive criticism.

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

**13**