## [Peer Review File · Royal Society Open Science]

Review History

RSOS-210474.R0 (Original submission)

Review form: Reviewer 1 (Ana Veríssimo)

Is the manuscript scientifically sound in its present form?

Yes

Are the interpretations and conclusions justified by the results?

Yes

Is the language acceptable?

Yes

Do you have any ethical concerns with this paper?

No

Have you any concerns about statistical analyses in this paper?

No

Recommendation?

Accept with minor revision (please list in comments)

Comments to the Author(s)

This paper is a valuable contribution to the field of taxonomy and highlights the great potential in using old specimens in museum wet-collections in clarifying some pending taxonomic questions. The paper is interesting to read, is scientifically sound and draws a simple but a valuable conclusion to Etmopterus taxonomy, but I think a change in the framing of the work would make the paper more appealing to a broader audience. Specifically, rather than asking one taxonomic question on one specific genus (and how the authors generated the data to answer it), the paper can be framed as presenting a simple methodological approach to answer pending taxonomic questions using valuable but underused museum specimens from old wet collections. Major changes would be needed mostly in the Introduction.

Minor changes/edits and comments were made in the pdf file of the submitted manuscript, in attachment (see Appendix A).

Review form: Reviewer 2

Is the manuscript scientifically sound in its present form?

Yes

Are the interpretations and conclusions justified by the results?

Yes

Is the language acceptable?

Yes

Do you have any ethical concerns with this paper?

No

Have you any concerns about statistical analyses in this paper?

No

Recommendation?

Major revision is needed (please make suggestions in comments)

Comments to the Author(s)

This paper reports the NGS sequencing of a syntype sample of *E. pusillus*, the extraction of mitochondrial sequences from the library of recovered sequences, the Sanger sequencing of NADH2 from a number of recently collected samples and evaluation of the NADH2 haplotype network for evidence to support the recently described species, *E. jounqi*.

The data collection is well done, although I have one concern (see below). The analysis of the 116 wet-preserved samples is impressive and the cost of 37 euros should allow other syntype samples to be sequenced. In fact, this low cost makes me wonder why only 1 syntype sample was included in this study.

My main concerns with the paper arise from the sample collection design to meet the stated objectives of the paper, the provenance and species ID of the samples included for the analysis, and relatively basic interpretation of the phylogeographic results. I will outline these below.

- 1) The title of the paper reads, “Sequencing of the mitochondrial genome of a 116 year old wet collection syntype confirms an almost global distribution of the smooth lantern shark”. I have problem with this title as I don’t see why the sequencing of a single sample would confirm the global distribution of a species? This point is never fully discussed in the paper. Yes, the syntypes NADH2 haplotype is within the diversity seen in samples collected more recently at various locations around the globe, but how does this syntype sample, in itself, confirm the Global distribution?
- 2) A number of syntypes for *E pusillus* exist (8 samples, line 94). Why was this particular samples chosen for analysis? Why not all 8 if you want to confirm the global distribution of the species? I presume this is because the sample was collected off Japan (closest geographically to the current *E joungi* samples), but this point is not address in the text. As a reader I have no context with which to judge the importance of this sample. For example, how does this sample relate to the study of Knuckey et al 2011, which describes *E joungi*? Does it clearly show it belongs to the description of *E pusillus* and not *E joungi*? It was collected about 100 years before Knuckey et al (2011) which analyzed samples collected north east of Taiwan (but actually from fish markets in Taiwan). Japan also lies north east of Taiwan. Do the records show exactly where the syntype was originally collected? Is there a possibility that the samples come from the same geographic population (East China Sea) of sharks?
- 3) In general, please provide more context for the samples used for the study. I understand that these samples are hard to come by and won’t represent the entire global range, but more effort should be made to show the distribution and sampling details of the samples. Table 1 is an unorganised list of the samples. This should be organized by species and then by geographic location. Samples new to this study should also be indicated. A number of samples/sequences were from Straube et al 2013. These should be indicated. Where possible the GPS (or relative) locations should be indicated, along with collection dates. I.e., how do the Tweed Heads, NSW samples differ from the Tasman Sea, NSW samples? Sample 1 is from the East China Sea. The Taiwan samples would also be from the East China Sea. Do these samples represent the different morphotypes from the same environment (more on the importance of this sample below)?
- 4) Following this point, and as species ID is really important for correct interpretation of phylogeographic patterns, please indicate how you determined the species ID of the samples in the North East Pacific. I.e., Was species determination conducted in light of the recent description of *E joungi*? For example, Sample 1 is listed as *E pusillus*, collected in the East China Sea and listed as sample number KU892588. In fact, the sample number is the Genbank accession number (noted as “in preparation”) for the whole mtDNA genome of a sample reported in Chen et al (2016). Chen et al described the sample provenance as “One specimen of *E. pusillus* was captured from continental shelf in East China Sea and landed on a pier in Wenling, Zhejiang, China. It was preserved in the museum of marine biology in Wenzhou Medical University with voucher WL2012051264.” No dates are given for the sample collection and consider that Wenling is much closer to Taiwan than Japan. Note, travelling west of Wenling on the East China Sea takes you to just northwest of Taiwan, around the same location where the samples used to describe *E joungi* were reported to be collected (Knuckley et al 2011). Note also, if this sample was collected prior to 2011, it would have, by default been listed as *E pusillus*. Please indicate how you determined it was actually *E pusillus* and not *E joungi*?
- 5) The haplotype network does not indicate which samples correspond to which haplotypes. Although it is relatively standard not to include labels in a network, I suggest that

you add the haplotype codes to Table 1 so that a comparison be made. Because I don't know which samples share haplotypes the following is part conjecture on my part. I noted that the syntype matches one of the North East Pacific samples. As the East China Sea haplotype (discussed in point 4) was used as the reference for the syntype reconstruction, can you please comment on whether these two samples differ at the NADH2 locus used for the phylogeographic comparison. If they differ great, but please note this in the results. However, if they match, I suspect there may be a bias associated with the reference in determining which mutations are real and which are artifacts. Six unique NADH2 haplotypes were found among seven recent North East Pacific samples (both species). Clearly more haplotypes are yet to be sampled in this geographic region. The odds that the syntype would match one already sampled are low. The odds that it would match the reference mtDNA are even lower. So, if they match, I am suspicious of a methodological bias in the determination of the syntype sequence.

Note, if these samples match, then they also match one of the *E. jounqi* samples. This is the only case of haplotype sharing in the North East Pacific. Speculation: if the East China Sea sample is actually another sample of *E. jounqi*, and there are doubts to the accuracy of the syntype haplotype, then you would need to reassess your conclusions.

6) I find the discussion too brief and the treatment of the evidence for or against the validity of *E. jounqi* mostly lacking. (Not having a good understanding of the provenance of the samples does not help my understand - covered above). I agree, the data as presented clearly does not provide support for *E. jounqi*. The reciprocal monophyly noted in many other groups does not exist here. However, the discussion should focus on the expectations of lineage sorting and not rely on a reference to a paper that this gene works well to sort out sharks in other genera. Based on the polyphyletic stage of the lineage sorting among two taxa examined, this haplotype network would suggest either an evolutionary recent speciation event or a single species. What is sum of evidence for either of these two hypotheses. Key here are the samples from the North East Pacific. Are both species types currently found East China Sea? Is this a regional morphotype of the same species or do the morphotypes overlap? Is there any evidence to suggest a lack of gene flow with the other locations? How do these findings relate to the samples used in the original description (Knuckey et al 2011)? Does Knuckey's paper also include other samples from the North East Pacific for its morphological comparison? Does it suggest the likely range distribution of the morphotype associated with *E. jounqi*? The entire East China Sea, or just around Taiwan? How does the sequence of the old (pre *E. jounqi* description) syntype relate to the interpretations? Does its original description fit the more recent description of *E. jounqi*, or is it more like *E. pusillus* samples used in the Knuckey paper? How does this all relate to the original description of *E. frontimaculatus* in this area? Lastly, what other studies could be done to resolve this question?

Minor points

1) A comparison with Straube et al (2013), shows three *E. jounqi* samples were analyzed in this earlier paper. The current study also uses three samples, two are the same but GN10140 is not used and a new sample, GN10148 is used instead. Given the paucity of data on this species, why would you exclude this previous sample? Or is this typo in Straube et al, Fig 2, or in the present table 1?

2) Figure 2 shows the distribution of NGS fragment sizes. The range is from 22 to 72+ bp and the average is reported as 30.5. This average is described in the results after the reporting of the final number of trimmed and filtered reads 8,970,773 (line 212). This is a bit confusing, as the trimmed sequences should not contain any reads below 30 bp (line 113/4). As Figure 2 appears to show the prefilter/trimmed reads, it should be reported prior to the reporting of the final

number of sequences used to assemble the mtDNA genome. Also, please explain what figure 2 is actually showing. Are these prefiltered or postfiltered reads?

- 3) Lines 296-301 describe the haplotype sharing observed among Pacific and Antarctic sleeper sharks. Is Edwards et al (2019) the correct reference? This paper is a review of the Greenland Shark. Is Murray et al (2008) the better reference?
- 4) Lines 58-59 state the *E bigelowi* is a more distant relative to *E pusillus* than *E jounqi*. What is the reference/evidence for this statement?
- 5) No samples of *E bigelowi* are included in this study. As another member of this "species group" these samples would likely help provide context for the interpretation of the *E jounqi* samples. If these are available I would suggest adding them to your haplotype network.
- 6) Please double check your reference list. A quick review found a number of typos/errors. i.e., line 390, inconsistent use of short and long forms for journals, Struabe et al 2015 is listed twice.

Decision letter (RSOS-210474.R0)

Dear Dr Straube

The Editors assigned to your paper RSOS-210474 "The mitochondrial genome of a 116 year old syntype confirms an almost global distribution of the Smooth Lantern Shark (Chondrichthyes: Etmopteridae)" have now received comments from reviewers and would like you to revise the paper in accordance with the reviewer comments and any comments from the Editors. Please note this decision does not guarantee eventual acceptance.

Both reviewers are positive about the work, but each raises substantive points that should be carefully considered and addressed. We invite you to respond to the comments supplied below and revise your manuscript. Below the referees' and Editors' comments (where applicable) we provide additional requirements. Final acceptance of your manuscript is dependent on these requirements being met. We provide guidance below to help you prepare your revision.

Please submit your revised manuscript and required files (see below) no later than 21 days from today's (ie 21-May-2021) date. Note: the ScholarOne system will 'lock' if submission of the revision is attempted 21 or more days after the deadline. If you do not think you will be able to meet this deadline please contact the editorial office immediately.

Please note article processing charges apply to papers accepted for publication in Royal Society Open Science (<https://royalsocietypublishing.org/rsos/charges>). Charges will also apply to papers transferred to the journal from other Royal Society Publishing journals, as well as papers

submitted as part of our collaboration with the Royal Society of Chemistry (<https://royalsocietypublishing.org/rsos/chemistry>). Fee waivers are available but must be requested when you submit your revision (<https://royalsocietypublishing.org/rsos/waivers>).

on behalf of Professor Marcelo Sanchez (Associate Editor) and Steve Brown (Subject Editor)
openscience@royalsociety.org

Reviewer comments to Author:
Reviewer: 1
Comments to the Author(s)

This paper is a valuable contribution to the field of taxonomy and highlights the great potential in using old specimens in museum wet-collections in clarifying some pending taxonomic questions. The paper is interesting to read, is scientifically sound and draws a simple but a valuable conclusion to Etmopterus taxonomy, but I think a change in the framing of the work would make the paper more appealing to a broader audience. Specifically, rather than asking one taxonomic question on one specific genus (and how the authors generated the data to answer it), the paper can be framed as presenting a simple methodological approach to answer pending taxonomic questions using valuable but underused museum specimens from old wet collections. Major changes would be needed mostly in the Introduction.

Minor changes/edits and comments were made in the pdf file of the submitted manuscript, in attachment.

Reviewer: 2
Comments to the Author(s)

This paper reports the NGS sequencing of a syntype sample of *E. pusillus*, the extraction of mitochondrial sequences from the library of recovered sequences, the Sanger sequencing of NADH2 from a number of recently collected samples and evaluation of the NADH2 haplotype network for evidence to support the recently described species, *E. jounqi*.

The data collection is well done, although I have one concern (see below). The analysis of the 116 wet-preserved samples is impressive and the cost of 37 euros should allow other syntype samples to be sequenced. In fact, this low cost makes me wonder why only 1 syntype sample was included in this study.

My main concerns with the paper arise from the sample collection design to meet the stated objectives of the paper, the provenance and species ID of the samples included for the analysis, and relatively basic interpretation of the phylogeographic results. I will outline these below.

1) The title of the paper reads, "Sequencing of the mitochondrial genome of a 116 year old wet collection syntype confirms an almost global distribution of the smooth lantern shark". I have problem with this title as I don't see why the sequencing of a single sample would confirm the

global distribution of a species? This point is never fully discussed in the paper. Yes, the syntypes NADH2 haplotype is within the diversity seen in samples collected more recently at various locations around the globe, but how does this syntype sample, in itself, confirm the Global distribution?

2) A number of syntypes for *E pusillus* exist (8 samples, line 94). Why was this particular samples chosen for analysis? Why not all 8 if you want to confirm the global distribution of the species? I presume this is because the sample was collected off Japan (closest geographically to the current *E joungi* samples), but this point is not address in the text. As a reader I have no context with which to judge the importance of this sample. For example, how does this sample relate to the study of Knuckey et al 2011, which describes *E joungi*? Does it clearly show it belongs to the description of *E pusillus* and not *E joungi*? It was collected about 100 years before Knuckey et al (2011) which analyzed samples collected north east of Taiwan (but actually from fish markets in Taiwan). Japan also lies north east of Taiwan. Do the records show exactly where the syntype was originally collected? Is there a possibility that the samples come from the same geographic population (East China Sea) of sharks?

3) In general, please provide more context for the samples used for the study. I understand that these samples are hard to come by and won't represent the entire global range, but more effort should be made to show the distribution and sampling details of the samples. Table 1 is an unorganised list of the samples. This should be organized by species and then by geographic location. Samples new to this study should also be indicated. A number of samples/sequences were from Straube et al 2013. These should be indicated. Where possible the GPS (or relative) locations should be indicated, along with collection dates. I.e., how do the Tweed Heads, NSW samples differ from the Tasman Sea, NSW samples? Sample 1 is from the East China Sea. The Taiwan samples would also be from the East China Sea. Do these samples represent the different morphotypes from the same environment (more on the importance of this sample below)?

4) Following this point, and as species ID is really important for correct interpretation of phylogeographic patterns, please indicate how you determined the species ID of the samples in the North East Pacific. I.e., Was species determination conducted in light of the recent description of *E joungi*? For example, Sample 1 is listed as *E pusillus*, collected in the East China Sea and listed as sample number KU892588. In fact, the sample number is the Genbank accession number (noted as "in preparation") for the whole mtDNA genome of a sample reported in Chen et al (2016). Chen et al described the sample provenance as "One specimen of *E. pusillus* was captured from continental shelf in East China Sea and landed on a pier in Wenling, Zhejiang, China. It was preserved in the museum of marine biology in Wenzhou Medical University with voucher WL2012051264." No dates are given for the sample collection and consider that Wenling is much closer to Taiwan than Japan. Note, travelling west of Wenling on the East China Sea takes you to just northwest of Taiwan, around the same location where the samples used to describe *E joungi* were reported to be collected (Knuckley et al 2011). Note also, if this sample was collected prior to 2011, it would have, by default been listed as *E pusillus*. Please indicate how you determined it was actually *E pusillus* and not *E joungi*?

5) The haplotype network does not indicate which samples correspond to which haplotypes. Although it is relatively standard not to include labels in a network, I suggest that you add the haplotype codes to Table 1 so that a comparison be made. Because I don't know which samples share haplotypes the following is part conjecture on my part. I noted that the syntype matches one of the North East Pacific samples. As the East China Sea haplotype (discussed in point 4) was used as the reference for the syntype reconstruction, can you please comment on whether these two samples differ at the NADH2 locus used for the phylogeographic comparison. If they differ great, but please note this in the results. However, if they match, I suspect there may be a bias associated with the reference in determining which mutations are real and which are artifacts.

Six unique NADH2 haplotypes were found among seven recent North East Pacific samples (both species). Clearly more haplotypes are yet to be sampled in this geographic region. The odds that the syntype would match one already sampled are low. The odds that it would match the reference mtDNA are even lower. So, if they match, I am suspicious of a methodological bias in the determination of the syntype sequence.

Note, if these samples match, then they also match one of the *E. jounqi* samples. This is the only case of haplotype sharing in the North East Pacific. Speculation: if the East China Sea sample is actually another sample of *E. jounqi*, and there are doubts to the accuracy of the syntype haplotype, then you would need to reassess your conclusions.

6) I find the discussion too brief and the treatment of the evidence for or against the validity of *E. jounqi* mostly lacking. (Not having a good understanding of the provenance of the samples does not help my understand - covered above). I agree, the data as presented clearly does not provide support for *E. jounqi*. The reciprocal monophyly noted in many other groups does not exist here. However, the discussion should focus on the expectations of lineage sorting and not rely on a reference to a paper that this gene works well to sort out sharks in other genera. Based on the polyphyletic stage of the lineage sorting among two taxa examined, this haplotype network would suggest either an evolutionary recent speciation event or a single species. What is sum of evidence for either of these two hypotheses. Key here are the samples from the North East Pacific. Are both species types currently found East China Sea? Is this a regional morphotype of the same species or do the morphotypes overlap? Is there any evidence to suggest a lack of gene flow with the other locations? How do these findings relate to the samples used in the original description (Knuckey et al 2011)? Does Knuckey's paper also include other samples from the North East Pacific for its morphological comparison? Does it suggest the likely range distribution of the morphotype associated with *E. jounqi*? The entire East China Sea, or just around Taiwan? How does the sequence of the old (pre *E. jounqi* description) syntype relate to the interpretations? Does its original description fit the more recent description of *E. jounqi*, or is it more like *E. pusillus* samples used in the Knuckey paper? How does this all relate to the original description of *E. frontimaculatus* in this area? Lastly, what other studies could be done to resolve this question?

Minor points

1) A comparison with Straube et al (2013), shows three *E. jounqi* samples were analyzed in this earlier paper. The current study also uses three samples, two are the same but GN10140 is not used and a new sample, GN10148 is used instead. Given the paucity of data on this species, why would you exclude this previous sample? Or is this typo in Straube et al, Fig 2, or in the present table 1?

2) Figure 2 shows the distribution of NGS fragment sizes. The range is from 22 to 72+ bp and the average is reported as 30.5. This average is described in the results after the reporting of the final number of trimmed and filtered reads 8,970,773 (line 212). This is a bit confusing, as the trimmed sequences should not contain any reads below 30 bp (line 113/4). As Figure 2 appears to show the prefilter/trimmed reads, it should be reported prior to the reporting of the final number of sequences used to assemble the mtDNA genome. Also, please explain what figure 2 is actually showing. Are these prefiltered or postfiltered reads?

3) Lines 296-301 describe the haplotype sharing observed among Pacific and Antarctic sleeper sharks. Is Edwards et al (2019) the correct reference? This paper is a review of the Greenland Shark. Is Murray et al (2008) the better reference?

4) Lines 58-59 state the *E. bigelowi* is a more distant relative to *E. pusillus* than *E. jounqi*. What is the reference/evidence for this statement?

5) No samples of *E. bigelowi* are included in this study. As another member of this “species group” these samples would likely help provide context for the interpretation of the *E. jounqi* samples. If these are available I would suggest adding them to your haplotype network.

6) Please double check your reference list. A quick review found a number of typos/errors. i.e., line 390, inconsistent use of short and long forms for journals, Struabe et al 2015 is listed twice.

===PREPARING YOUR MANUSCRIPT===

===PREPARING YOUR REVISION IN SCHOLARONE===

Author's Response to Decision Letter for (RSOS-210474.R0)

See Appendix B.

RSOS-210474.R1 (Revision)

Review form: Reviewer 1 (Ana Veríssimo)

Is the manuscript scientifically sound in its present form?

Yes

Are the interpretations and conclusions justified by the results?

Yes

Is the language acceptable?

Yes

Do you have any ethical concerns with this paper?

No

Have you any concerns about statistical analyses in this paper?

No

Recommendation?

Accept with minor revision (please list in comments)

Comments to the Author(s)

Dear Authors,

I have read the revised version of the MS and I believe the changes incorporated in the text following the two reviewer's suggestions have improved the quality of the results and of the MS overall. I have only two minor comments to make:

1. In line with the broader scope of the MS now, i.e. how generating reference sequence data from old, wet-collection specimens from Museum collections is important and can be made following your approach, I believe you can further reinforce the relevance of the paper by highlighting that you successfully generated key reference genetic sequence data for type specimens -which are key in solving some pending taxonomic questions, regarding for example taxonomic validity of previous synonymy. Indeed, you could have reinforced the fact that the data supported the synonymy of *E. frontimaculatus* as *E. pusillus* proposed by Shirai & Tachikawa (1993) based on morphological data.
2. The symbols and colors of the new Figure 1 (map) could be changed to improve readability. For instance, you could use different symbols per species (e.g. circles, squares, diamonds). As is, you have 4 different colors to designate the same species, although the colors denote different sampling locations (not indicated in the legend but implied from the map). I would also include the reference of "*Etmopterus pusillus* syntype" to the top of the remainder *E. pusillus* references, or simply ordering the color/symbol legend alphabetically. If possible, the same symbols and colors should be consistent between the new Figures 1 and 6.

Review form: Reviewer 2

Is the manuscript scientifically sound in its present form?

Yes

Are the interpretations and conclusions justified by the results?

Yes

Is the language acceptable?

Yes

Do you have any ethical concerns with this paper?

No

Have you any concerns about statistical analyses in this paper?

No

Recommendation?

Accept with minor revision (please list in comments)

Comments to the Author(s)

I find the revised manuscript to be significant improvement from the first submission and am generally happy with the authors responses to my initial concerns that mostly arose from the lack of clarity of results. I agree with the revised title and feel the tables and figures (especially Fig 6) have been much improved. I only have a have few minor points for consideration.

Minor points.

1) Word usage, mostly in the revised text, could be improved through another round of edits. The first sentence of the abstract is awkwardly written. See also: page 3, last sentence of second paragraph; page 3, first sentence of third paragraph; page 11, consider replace the word "However" which starts the first paragraph with "Despite the observed DNA damage".

2) The replacement of "Northeast" for "North East", etc. was only partially completed in the revised manuscript.

3) I am not sure your removal of authors names in the text and replacement with bracketed references is correct. E.g., on page 4 - "In 2011, however, (18) described a morphological similar species ...", I think this should read "In 2011, however, Knuckey et al. (18) described a morphological similar species ..." A scan of recent articles in this journal shows the later usage. Which reads a lot nicer - :-).

4) page 12. I don't find the end of the second last paragraph clear. The logic sounds like your conclusion that the morphological differences noted for E joungi must be part of the wider intraspecific variation is what leads to your recommendation that E joungi is a junior synonym. This recommendation is actually based solely on the mtDNA results, which leads to a reinterpretation of Knuckley's results. Consider revising. E.g., The expectation of monophyly is not met for E. joungi. ... Our results suggest that the detected morphological differences noted between E joungi and the comparator E pusillus samples (18) represent the range of intraspecific variation in this widespread species and that E joungi is a junior synonym of E pusillus. Or something to this effect. Perhaps I am overthinking this, but the way it was written didn't sound quite right to me.

Decision letter (RSOS-210474.R1)

Dear Dr Straube

On behalf of the Editors, we are pleased to inform you that your Manuscript RSOS-210474.R1 "Mitochondrial DNA sequencing of a wet-collection syntype demonstrates the importance of type material as genetic resource for Lantern Shark taxonomy" has been accepted for publication in Royal Society Open Science subject to minor revision in accordance with the referees' reports. Please find the referees' comments along with any feedback from the Editors below my signature.

Please submit your revised manuscript and required files (see below) no later than 7 days from today's (ie 11-Aug-2021) date. Note: the ScholarOne system will 'lock' if submission of the revision is attempted 7 or more days after the deadline. If you do not think you will be able to meet this deadline please contact the editorial office immediately.

on behalf of Professor Marcelo Sanchez (Associate Editor) and Steve Brown (Subject Editor)
openscience@royalsociety.org

Reviewer comments to Author:
Reviewer: 1
Comments to the Author(s)

Dear Authors,

I have read the revised version of the MS and I believe the changes incorporated in the text following the two reviewer's suggestions have improved the quality of the results and of the MS overall. I have only two minor comments to make:

1. In line with the broader scope of the MS now, i.e. how generating reference sequence data from old, wet-collection specimens from Museum collections is important and can be made following your approach, I believe you can further reinforce the relevance of the paper by highlighting that you successfully generated key reference genetic sequence data for type specimens -which are key in solving some pending taxonomic questions, regarding for example taxonomy validity of previous synonymy. Indeed, you could have reinforced the fact that the data supported the synonymy of *E. frontimaculatus* as *E. pusillus* proposed by Shirai & Tachikawa (1993) based on morphological data.

2. The symbols and colors of the new Figure 1 (map) could be changed to improve readability. For instance, you could use different symbols per species (e.g. circles, squares, diamonds). As is, you have 4 different colors to designate the same species, although the colors denote different sampling locations (not indicated in the legend but implied from the map). I would also include the reference of "*Etmopterus pusillus* syntype" to the top of the remainder *E. pusillus* references, or simply ordering the color/symbol legend alphabetically. If possible, the same symbols and colors should be consistent between the new Figures 1 and 6.

Reviewer: 2

Comments to the Author(s)

I find the revised manuscript to be significant improvement from the first submission and am generally happy with the authors responses to my initial concerns that mostly arose from the lack of clarity of results. I agree with the revised title and feel the tables and figures (especially Fig 6) have been much improved. I only have a few minor points for consideration.

Minor points.

1) Word usage, mostly in the revised text, could be improved through another round of edits. The first sentence of the abstract is awkwardly written. See also: page 3, last sentence of second paragraph; page 3, first sentence of third paragraph; page 11, consider replace the word "However" which starts the first paragraph with "Despite the observed DNA damage".

2) The replacement of "Northeast" for "North East", etc. was only partially completed in the revised manuscript.

3) I am not sure your removal of authors names in the text and replacement with bracketed references is correct. E.g., on page 4 - "In 2011, however, (18) described a morphological similar species ...", I think this should read "In 2011, however, Knuckey et al. (18) described a morphological similar species ..." A scan of recent articles in this journal shows the later usage. Which reads a lot nicer - :-).

4) page 12. I don't find the end of the second last paragraph clear. The logic sounds like your conclusion that the morphological differences noted for *E. joungi* must be part of the wider intraspecific variation is what leads to your recommendation that *E. joungi* is a junior synonym. This recommendation is actually based solely on the mtDNA results, which leads to a reinterpretation of Knuckley's results. Consider revising. E.g., The expectation of monophyly is not met for *E. joungi*. ... Our results suggest that the detected morphological differences noted between *E. joungi* and the comparator *E. pusillus* samples (18) represent the range of intraspecific variation in this widespread species and that *E. joungi* is a junior synonym of *E. pusillus*. Or something to this effect. Perhaps I am overthinking this, but the way it was written didn't sound quite right to me.

===PREPARING YOUR MANUSCRIPT===

===PREPARING YOUR REVISION IN SCHOLARONE===

- An individual file of each figure (EPS or print-quality PDF preferred [either format should be produced directly from original creation package], or original software format).
- An editable file of each table (.doc, .docx, .xls, .xlsx, or .csv).
- An editable file of all figure and table captions.

- Any electronic supplementary material (ESM).
- If you are requesting a discretionary waiver for the article processing charge, the waiver form must be included at this step.
- If you are providing image files for potential cover images, please upload these at this step, and inform the editorial office you have done so. You must hold the copyright to any image provided.
- A copy of your point-by-point response to referees and Editors. This will expedite the preparation of your proof.

- Ensure that your data access statement meets the requirements at <https://royalsociety.org/journals/authors/author-guidelines/#data>. You should ensure that you cite the dataset in your reference list. If you have deposited data etc in the Dryad repository, please only include the 'For publication' link at this stage. You should remove the 'For review' link.
- If you are requesting an article processing charge waiver, you must select the relevant waiver option (if requesting a discretionary waiver, the form should have been uploaded at Step 3 'File upload' above).
- If you have uploaded ESM files, please ensure you follow the guidance at <https://royalsociety.org/journals/authors/author-guidelines/#supplementary-material> to include a suitable title and informative caption. An example of appropriate titling and captioning may be found at https://figshare.com/articles/Table_S2_from_Is_there_a_trade-off_between_peak_performance_and_performance_breadth_across_temperatures_for_aerobic_sc_ope_in_teleost_fishes_/3843624.

Author's Response to Decision Letter for (RSOS-210474.R1)

See Appendix C.

Decision letter (RSOS-210474.R2)

Dear Dr Straube,

I am pleased to inform you that your manuscript entitled "Mitochondrial DNA sequencing of a wet-collection syntype demonstrates the importance of type material as genetic resource for Lantern Shark taxonomy" is now accepted for publication in Royal Society Open Science.

on behalf of Professor Marcelo Sanchez (Associate Editor) and Steve Brown (Subject Editor)
openscience@royalsociety.org

Appendix A**ROYAL SOCIETY
OPEN SCIENCE****The mitochondrial genome of a 116 year old syntype
confirms an almost global distribution of the Smooth
Lantern Shark
(Chondrichthyes: Etmopteridae)**

Journal:	Royal Society Open Science
Manuscript ID	RSOS-210474
Article Type:	Research
Date Submitted by the Author:	23-Mar-2021
Complete List of Authors:	Straube, Nicolas; University Museum of Bergen, Advelingen for naturhistorie Preick, Michaela; University of Potsdam, Institut für Biochemie und Biologie Naylor, Gavin; University of Florida, Florida Museum of Natural History Hofreiter, Michael; University of Potsdam, Institute for Biochemistry and Biology
Subject:	molecular biology < BIOLOGY, taxonomy and systematics < BIOLOGY
Keywords:	museum preserved specimens, archival DNA, deep-sea sharks, Etmopterus pusillus, Etmopterus jounji, taxonomy
Subject Category:	Genetics and genomics

Author-supplied statements

Relevant information will appear here if provided.

Ethics

Does your article include research that required ethical approval or permits?:

This article does not present research with ethical considerations

Statement (if applicable):

CUST_IF_YES_ETHICS :No data available.

Data

It is a condition of publication that data, code and materials supporting your paper are made publicly available. Does your paper present new data?:

Yes

Statement (if applicable):

Trimmed sequencing reads, mitochondrial genome sequence of the *Etmopterus pusillus* syntype as well as the NADH2 alignment for the network analysis are available at <https://datadryad.org/stash/share/NhiU9vM-OoEqCrhG0BxJccSIQGTWv59Yop6petDjto0> for revisions.

Conflict of interest

I/We declare we have no competing interests

Statement (if applicable):

CUST_STATE_CONFLICT :No data available.

Authors' contributions

This paper has multiple authors and our individual contributions were as below

Statement (if applicable):

NS and MH designed the study. NS and MP conducted laboratory steps. NS and MH analysed the data. GJP provided NADH2 sequences for the network analysis. NS wrote and drafted the manuscript with contributions from all authors. The final version of the manuscript was approved by all authors.

**Sequencing the mitochondrial genome of a 116 year old wet collection syntype confirms an**
**almost global distribution of the Smooth Lantern Shark**
**(Chondrichthyes: Etmopteridae)**

Nicolas Straube^{1*}, Michaela Preick², Gavin J. P. Naylor³ & Michael Hofreiter²

¹ University Museum of Bergen, Department of Natural History, Allégaten 41, 5007 Bergen, Norway

² University of Potsdam, Evolutionary and Adaptive Genomics, Karl-Liebknecht-Str. 24-25, 14476 Potsdam, Germany

³ Florida Museum of Natural History, University of Florida, Cultural Plaza 3215 Hull Road, Gainesville, FL 32611-2710, USA

* corresponding author: nicolas.straube@uib.no

**Abstract**

Using an ancient DNA extraction protocol, we recovered DNA from a 116 year old syntype specimen
of the Smooth Lanternshark, *Etmopterus pusillus*. The DNA was converted into a single stranded
DNA library and shotgun sequenced. Approximately 9 million reads were used for analyses on DNA
characteristics and reconstruction of the mitochondrial genome. The archival DNA is highly
fragmented, comparable to conditions in ancient DNA samples. A total of 4,599 mitochondrial reads
were available for the genome reconstruction using an iterative mapping approach. The resulting
genome sequence has a 12 times coverage and a length of 16,741 basepairs. All 37 vertebrate
mitochondrial loci plus the control region were identified and annotated. The mitochondrial NADH2
gene was subsequently used to place the syntype haplotype in a network comprising multiple *E.*
*pusillus* samples from various distant localities as well as sequences from a morphological similar
species, the Shortfin Smooth Lantern Shark *Etmopterus jounqi*. Results confirm the almost global
distribution of *E. pusillus* and suggest *E. jounqi* to be a junior synonym of *E. pusillus*. As
mitochondrial DNA often represents the only available reference information in non-model
organisms, this study illustrates how mitochondrial DNA from an aged, wet-collection type specimen
can be cost-efficiently sequenced.

**Keywords**

museum preserved specimens; archival DNA; deep-sea sharks; *Etmopterus pusillus*; *Etmopterus*
*jounqi*; taxonomy

Introduction

Lantern sharks (Etmopteridae) are the most species-rich shark family with 53 species described in 4 genera (Pollerspöck & Straube 2020). They occur predominantly in depths of 200 to 1000 meters and are distributed globally with the exception of the North East Pacific (Ebert et al. 2013). Lantern sharks reach sizes of little more than 100 cm and include some of the smallest known shark species, for example *E. perryi*, which matures at about 18 cm (Ebert et al. 2013). The species-rich genus *Etmopterus*, containing 44 species (Pollerspöck & Straube 2020), is subdivided into four monophyletic subgroups: The *E. spinax*, *E. gracilispinis*, *E. pusillus* and *E. lucifer* clades (Straube et al. 2010).

Despite their worldwide distribution and species diversity, their phylogenetic relationships have not been studied extensively at the species level. Exact distribution ranges are often poorly known due to the scarcity of specimens available, or a lack of comparative studies that incorporate samples from large geographic areas. While some species appear localized, such as *E. lailae* Ebert, Papastamatiou, Kajiura & Wetherbee, 2017 or *E. marshae* Ebert & Van Hees, 2018, others are widespread, for example *E. granulosus* (Günther 1880) or *E. viator* Straube, 2011 (Straube et al. 2011a, 2011b, 2015). Given morphological similarities between closely related species and a lack of documentation of morphological changes throughout ontogeny and/ or differences between the sexes, the geographical origin of a sample is often relied upon to distinguish among congeners that are morphologically similar.

The Smooth Lanternshark, *Etmopterus pusillus* (Lowe, 1839) typifies the challenge outlined above. As its common name indicates, it is characterized by smooth skin, a feature caused by block-like rather than the typical tooth-like dermal denticles, a rare feature among etmopterids in general and within the *E. pusillus* clade only shared by its sister taxon *E. joungi* Knuckey et al. 2011 and the more distantly related Blurred Smooth Lanternshark, *Etmopterus bigelowi* Shirai & Tachikawa 1993. The Smooth Lanternshark was originally described from Madeira in the North East Atlantic Ocean. After its description in 1839, morphologically similar specimens were reported from numerous geographically distant locations far from the type locality suggesting the species was widespread. Shirai & Tachikawa (1993) were the first to explore the taxonomic status of the morphologically similar specimens. These authors described a new species, *E. bigelowi* and commented on the distribution range of *E. pusillus* as occurring almost globally by declaring the North West Pacific species *E. frontimaculatus* Pietschmann 1907 as a junior synonym to *E. pusillus*. In 2011, however, Knuckey et al. described a morphologically similar species from the North West Pacific (Taiwan), the Shortfin Smooth Lanternshark *E. joungi*. The species is described to occur localized off Taiwan.

2

Later studies based on the analyses of mitochondrial NADH dehydrogenase subunit 2 gene sequences
showed that mitochondrial haplotypes of *E. pusillus* from both the Pacific and Atlantic Ocean mix
along with specimens identified as *E. joungi* from Taiwan (Straube et al. 2013). The authors suggested
that the species status of *E. joungi* be reviewed to clarify whether *E. joungi* represented a sub-
population of *E. pusillus* or indeed a distinct species. Here, we applied an ancient DNA approach to
sequence archival mitochondrial DNA of one of the syntypes of *E. pusillus* collected in 1903 in the
75 North West Pacific off Japan, which represents a specimen of the former type series of *E.*
*frontimaculatus*.

Recent methodological advances in accessing archival DNA from museum wet collections
show that archival DNA is accessible by the application of ancient DNA methods (Straube et al. in
revision; Lyra et al. 2020, Rancilhac et al. 2020). In this study, we reconstruct the full mitochondrial
genome of the 116 year old ethanol preserved type specimen with an unknown fixation history from
shotgun short read sequences. Further, we use sequence information from the mitochondrial NADH
dehydrogenase subunit 2 gene (NADH2) for placing the syntype haplotype in a phylogenetic network
of a broader sampling of *E. pusillus* and *E. joungi* mitochondrial NADH2 sequences. We use the
sequence information to estimate the haplotype diversity from both species and discuss the
85 distribution range of *E. pusillus*.

32 33 **2. Material & Methods**

34 35 36 2.1 Mitochondrial genome sequencing of the *E. pusillus* syntype museum specimen

Tissue from the syntype was sampled during a research visit by NS to the Museum of Natural History
(MNH) in Vienna, Austria in 2017. The syntype series comprises eight specimens cataloged with
three collection lot numbers, i.e., NMW-78526, NMW-65832, NMW-61473. The tissue sample
analyzed herein was dissected from specimen NMW-78526-15. We removed 67,7 mg of muscle
tissue with a sterile scalpel and tweezers and kept it in the original preservation fluid until DNA
extraction. The specimen was originally collected in 1903. By the time of DNA extraction in 2019, it
was therefore 116 years old. Details on its fixation and preservation history are unknown, but we
assume that it was preserved in the same way as during our tissue sampling, i.e., as a wet collection
specimen in supposedly diluted ethanol since it arrived at the museum.

Due to the age and preservation of the syntype specimen, we applied an ancient DNA approach
for sequencing mitochondrial DNA. As Straube et al. (in revision) show, DNA fragmentation is
usually advanced with average fragments lengths of 50 bp in museum wet collection specimens and

the application of ancient DNA extraction methods is thus appropriate for recovering DNA from such
specimens. Therefore, we followed the procedure described in Straube et al. (in revision) using the
Guanidine treatment DNA extraction approach modified from (Dabney et al. 2013a; Rohland et al.
2004). The full amount of tissue (67,7 mg) was used for the extraction. Thereafter, we converted the
DNA fragments into a single-stranded DNA library following the protocol by Gansauge & Meyer
(2013). Library fragment size was checked using an Agilent TapeStation®. DNA extraction and
library preparation were conducted in a dedicated historical DNA laboratory at the University of
110 Potsdam following the recommendations in Fulton & Shapiro (2019). The library was test-sequenced
on an Illumina NextSeq® instrument with 1,758,548 75 bp single-end reads using the 500/550 High
Output v2.5 (75 cycles, Illumina 20024906) kit and custom primers (Paijmans et al. 2017). Reads
were quality filtered and trimmed using cutadapt (Martin 2011). We defined a quality cut-off of 20
and a minimum read length of 30 bp. Presence of target DNA was checked by using Fastqscreen
(Wingett & Andrews 2018). Phylogenetically distant reference genomes to check for the level of
contamination were *Homo sapiens* (Wingett & Andrews 2018), *Escherichia coli* (Wingett & Andrews
2018), *Rhincodon typus* (Read et al. 2017), *Danio rerio* (Howe et al. 2013), *Anolis carolinensis* (Alfödi
et al. 2011) and *Xenopus tropicalis* (Karimi et al. 2018; Wyffels et al. 2013). Further, a set of
115 mitochondrial genomes, vector and adapter sequences provided by Fastqscreen was included in the
analysis (Wingett & Andrews 2018). The *Etmopterus spinax* (Linnaeus, 1758) transcriptome from
Delroisse et al. (2019) and the *E. pusillus* mitochondrial genome from Chen et al. (2016) were used
as target references.

Based on the fastqscreen analysis results, we started an initial experiment with Mitobim (Hahn,
Bachmann & Chevreur, 2013), where we first assembled reads using MIRA (Chevreur, Wetter &
Suhai, 1999) and then initiated an iterative mapping approach defining the *E. pusillus* mitochondrial
genome (KU892588) as seed for mapping. Subsequently, we counted the number of reconstructed
sites in the last iteration. Then we divided the number of recovered mitochondrial nucleotides by the
total number of reads from test sequencing resulting in the theoretical number of mitochondrial
nucleotides per read. This allowed us to predict the number of further reads necessary to be able to
reconstruct the full mitochondrial genome sequence guiding subsequent deeper sequencing of the
DNA library.

Based on the estimated number of reads necessary, we targeted a total of 10 million reads for
sufficient coverage of the mitochondrial genome. Reads were quality filtered and trimmed as
described above for the test-sequencing dataset. For estimating DNA damage pattern in the sample,
BWA (Li & Durbin 2009) was used to map reads to the *Etmopterus spinax* transcriptome reference
derived from Delroisse et al. (2019). Samtools (Li et al. 2009) was subsequently used to remove reads

with low mapping quality (-q 30) and potential PCR duplicates (rmdup). The resulting bam file was
input to an analysis with Mapdamage vers. 2.0.7 (Johnson et al. 2013), which was run under default
settings to estimate the damage patterns of the archival syntype DNA.

[revised manuscript text omitted]

PopART vers. 1.7 (Leigh & Bryant 2015) was used to reconstruct a haplotype network from
the NADH2 sequences. PopART was run using the Median Joining inference under default settings
providing a trait file coding for locality information.

3. Results

3.1 Syntype archival DNA

We extracted 204 ng of DNA from an initial 67.7 mg of muscle tissue. 163,2 ng were used for
construction of an Illumina sequencing library using the single stranded approach from (Gansauge &
Meyer 2013). After adapter ligation, qPCR indicated an optimal number of 15 cycles to amplify the
library in the indexing step. The library concentration was 8.54 ng/µl with a size peak measured with
the TapeStation® at 168 bp. A total of 1,758,548 reads were available after test sequencing. After
200 trimming, 782,282 reads were used for checking presence of target DNA. The Fastqscreen analysis

subsampled 99,898 reads and detected 4.53 % unique hits (4,526 reads) mapping to the *E. spinax*
transcriptome and 0.07 % unique hits (73 reads) mapping to the mitochondrial genome (Figure 1).

Using the test sequencing data for the Mitobim analysis, 11,122 mitochondrial nucleotides were
recovered after 13 iterations. Assuming a total mitochondrial genome sequence length of 16,729 bases
(equaling the length of the reference mitochondrial genome KU892588 (Table 1)), we targeted
13,940,833 reads for a fivefold coverage of the mitochondrial genome. ~~Additional~~ 9,901,517 reads
were subsequently sequenced. After trimming, 8,188,491 reads were available for combination with
the trimmed test sequencing reads equaling a total of 8,970,773 trimmed reads for reconstruction of
the mitochondrial genome and further analysis. The distribution of sequencing reads shows a high
level of fragmentation with the majority of reads below a length of 50 bp (Figure 2). The average
insert size of the library is 30.5 nucleotides.

The Mapdamage analysis shows an increased C to T substitution rate at the 3' end likely documenting
cytosine deamination. The 3' end additionally shows an elevated rate of A to G substitutions
indicating depurination (Dabney et al. 2013b). Slightly elevated rates are also documented for the 5'
end (Figure 3).

3.2 Syntype mitochondrial DNA and genome reconstruction

After assembling trimmed reads with Mitobim, the full mitochondrial genome was recovered from a
total of 4,559 trimmed and quality filtered mitochondrial reads after three iterations. The analysis
resulted in a single consensus sequence of 16,741 bp with an average per base coverage of 12.12
(Table 2, Figure 4). Mapping reads used for mitochondrial genome reconstruction by Mitobim back
to the reconstructed syntype mitochondrial genome show that 3,808 (81.8 %) of 4654 reads map.
80.5 % of reads have a mapping quality of ≥ 30 . Mapping all available 8,188,491 trimmed reads
derived from shotgun sequencing onto the reconstructed mitochondrial genome results in a similar
number (3,850) of reads mapping to the reconstructed genome sequence.

Estimated damage patterns of the archival mitochondrial DNA show an elevated rate of cytosine to
thymine substitutions at the 5' end of reads indicating cytosine deamination. The level of deamination
is below 5 %. It is around 10 % at the 3' end, however (Figure 3B). Annotation of the mitochondrial
genome with Geneious allowed all coding genes, 12S rRNA, 16S rRNA and tRNAs to be identified,
as well as defining the control region. Table 2 shows the summary statistics for the mitochondrial
genome sequence. Based on the reference annotation of Chen et al. (2016), we confirm that the
syntype mitochondrial genome comprises the complete set of 37 vertebrate mitochondrial genes, i.e.
13 protein-coding genes, two ribosomal RNAs, and 22 transfer RNAs as well as the control region.

The majority of loci was encoded on the H strand except for eight tRNAs (trnQ, trnA, trnN, trnC,
trnY, trnS2, trnE, trnP) and a single protein coding gene, NAD6. The sequence of loci is shown in
Figure 5.

3.3 Haplotype network

After manual editing of the NADH2 alignment in Geneious, 1043 nucleotides each of 37 samples in
total were available for the analysis (Table 1). PopART detected 25 haplotypes with a nucleotide
diversity π of 0.004. There are 35 segregating sites and 18 parsimony informative characters. The
network visualized in Figure 6 shows two closely related haplotype groups, one comprising
haplotypes from the North East Atlantic as well as the South West Pacific, while the other one
includes haplotypes from the North and South West Pacific including the syntype of *E. pusillus* and
*E. jounqi* specimens. Both groups are separated by two mutational steps. The syntype haplotype is
identical to an *E. jounqi* haplotype as well as a North West Pacific *E. pusillus* haplotype (Figure 6).

4. Discussion

4.1 Effects of long-term storage on archival syntype DNA of *E. pusillus*

As shown in Figure 2, the fragmentation level of the archival DNA recovered is high, with an average
fragment length of 30.5 bp indicating substantial degradation of the DNA. Indeed, the preservation
and potential but hitherto unknown fixation of this sample led within a mere 116 years to a state of
DNA fragmentation comparable to that of ancient DNA (Figure 3A; Dabney et al. 2013b). The exact
causes of this drastic and relatively rapid degradation remain unknown, but if the specimen had been
fixed in formaldehyde, cross-linking of DNA strands or DNA to proteins as well as single strand
breaks have likely occurred. The preservation in diluted EtOH over more than a century almost
265 certainly also caused DNA hydrolysis (Straube et al., in revision). Deamination levels seem lower
than in ancient DNA samples, but this is not unexpected, given that DNA deamination has been
shown to be at least partially time-dependent (Sawyer et al. 2012).

However, the ancient DNA extraction method applied, which is optimized for the recovery of
270 single stranded DNA library. The fastqscreen analysis indicated the presence of target DNA, but the
majority of reads could not be assigned to any of the given reference sequences indicating a high level
of microbial contamination. Although this would argue in favor of further processing the library with

target enrichment approaches to capture target DNA sequences of interest, the initial mapping
experiment with Mitobim showed that enough mitochondrial reads should be recovered with only 10
million reads corresponding to sequencing costs of only 37, - € in the dedicated historical laboratory
of the University of Potsdam. Recovering the full mitochondrial genome from shotgun sequencing
saves time-consuming and costly target capture approaches which demand additional laboratory
of working time besides prior bait design and production. Test sequencing of libraries using a limited
number of reads as a first step to check for target DNA and contamination levels has been successfully
applied as adequate strategy for sequencing dry museum and herbarium specimens (e.g., Zeng et al.
2018). Following previous studies, we confirm this strategy to be useful for our wet-collection sample
as well.

4.2 Distribution of *Etmopterus pusillus* and taxonomic implications

Within Etmopteridae, large distribution ranges are known for several species (e.g., Straube et al.
2011a, b; 2015). Our results support the notion by Shirai and Tachikawa (1993) that *E. pusillus* is a
wide-spread species. In fact, it is distributed almost circumglobally. The lack of molecular differences
across vast geographical distances is striking (Figure 6). Indeed, South West Pacific mitochondrial
haplotypes cluster together with both North East Atlantic (close to the type locality of *E. pusillus*)
and North West Pacific haplotypes. Substantial long distance dispersal potential has been suggested
as an explanation to account for the lack of population structure in several deep-sea shark species
(Murray et al. 2008; Veríssimo et al. 2011; Walter et al. 2017) including *Etmopterus* species (Straube
et al. 2011a).

Mitochondrial sequence divergence is often used for species delimitation in sharks (e.g., Naylor
et al. 2012a). Admixture of mitochondrial haplotypes is therefore impacting taxonomy. North Pacific
mitochondrial haplotypes of the Pacific Sleeper Shark *Somniosus pacificus* Bigelow & Schröder,
1944 are also found in specimens assigned to the Antarctic Sleeper Shark *Somniosus antarcticus*
Whitley, 1939, for example, which questions the taxonomic status of *S. antarcticus* (Edwards et al.
2019; Yano et al. 2004). The results presented herein also impact the taxonomy of *E. jounqi*. Our
haplotype network shows that the *E. pusillus* syntype shares a haplotype with a specimen identified
previously as *E. jounqi* in Straube et al. (2013) making a distinction of the two species with
mitochondrial NADH2 sequence data impossible. This mitochondrial gene sequence, however, was
demonstrated to be highly diagnostic for species level distinction in chondrichthyan fishes (Naylor et
al. 2012a; Straube et al. 2013). We therefore suggest that *E. jounqi* is a junior synonym of *E. pusillus*.

Even though etmopterids are not directly targeted in fisheries, the shift of commercial fisheries to fishing grounds in deeper waters (Morato et al. 2006; Rogers & Gianni 2010) poses a threat to slow-growing and late maturing species with few offspring such as *E. pusillus*. Detailed information on the distribution ranges of such species is essential for precise taxonomic assessment and is the basis for correct evaluation for protection and management strategies. With regard to *E. pusillus*, our results of a global marine distribution give hope for cautious optimism regarding its future. Since little is known about geographical ranges and population sizes of many shark species, especially deep sea ones, filling these gaps in our knowledge is of prime importance, as only then, conservation efforts can be focused on those species that are under the greatest danger of extinction.

References

- Alföldi, J, et al. 2011 The genome of the green anole lizard and a comparative analysis with birds and mammals. *Nature* **477**(7366), 587–591. (doi: 10.1038/nature10390)
- Bernt M, Donath A, Jühling F, Externbrink F, Florentz C, Fritzsche G, Pütz J, Middendorf M, Stadler PF 2013 MITOS: Improved de novo Metazoan Mitochondrial Genome Annotation. *Mol Phylogenet Evol* **69** (2), 313–319.
- Chen H, Chen X, Gu X, Wan H, Chen X, Ai W 2016 The phylogenomic position of the smooth lanternshark *Etmopterus pusillus* (Squaliformes: Etmopteridae) inferred from the mitochondrial genome. *Mitochondrial DNA Part B* **1**(1), 341-342, (doi: 10.1080/23802359.2016.1172274)
- Chevreur B, Wetter T, Suhai S 1999 Genome Sequence Assembly Using Trace Signals And Additional Sequence Information. *Computer Science and Biology: Proceedings of the German Conference on Bioinformatics (GCB)* **99**, 45–56.
- Dabney J, Knapp M, Glock I, Gansauge M, Weihmann A, Nickel B, ... Meyer M 2013a. Complete mitochondrial genome sequence of a Middle Pleistocene cave bear reconstructed from ultra short DNA fragments. *P Natl a Sci* **110**(39), 15758-15763. (doi: 10.1073/pnas.1314445110)
- Dabney J, Meyer M, Pääbo S 2013 Ancient DNA damage. *Cold Spring Harbor perspectives in biology* **5**(7), a012567. (doi: https://doi.org/10.1101/cshperspect.a012567)
- Delroisse J et al. 2019 De novo transcriptome analyses provide insights into opsin-based photoreception in the lanternshark *Etmopterus spinax*. *PLOS ONE* **13**(12), E0209767.
- Ebert DA, Fowler S, Compagno LJV 2013 *Sharks of the World: A Fully Illustrated Guide*.
- Ebert DA, Papastamatiou YP, Kajiura SM, Wetherbee BM 2017 *Etmopterus lailae* sp. nov., a new lanternshark (Squaliformes: Etmopteridae) from the Northwestern Hawaiian Islands.

*Zootaxa* **4237** (2), 371–382. (doi: [10.11646/zootaxa.4237.2.10](https://doi.org/10.11646/zootaxa.4237.2.10))
- Ebert DA, Van Hees KE 2018 *Etmopterus marshae* sp. nov, a new lanternshark (Squaliformes:
Etmopteridae) from the Philippine Islands, with a revised key to the *Etmopterus lucifer* clade.
*Zootaxa* **4508** (2), 197–210. (doi: [10.11646/zootaxa.4508.2.3](https://doi.org/10.11646/zootaxa.4508.2.3))
- Edwards JE et al 2019 Advancing Research for the Management of Long-Lived Species: A Case
Study on the Greenland Shark. *Frontiers in Marine Science* **6**, 1–87.
(doi:10.3389/fmars.2019.00087)
- Fulton TL, Shapiro B 2019 Setting Up an Ancient DNA Laboratory. In: Shapiro B, Barlow A,
Heintzman P, Hofreiter M, Paijmans J, Soares A (eds) Ancient DNA. Methods in Molecular
Biology, vol 1963 (pp.1–13). Humana Press, New York, NY. (doi: 10.1007/978-1-4939-9176-
1_1)
20 350
- Gansauge MT, Meyer M 2013 Single-stranded DNA library preparation for the sequencing of ancient
or damaged DNA. *Nat Protoc*, **8** (4), 737–748. (doi: 10.1038/nprot.2013.038)
- Hahn C, Bachmann L, Chevreux B 2013 Reconstructing mitochondrial genomes directly from
genomic next-generation sequencing reads—a baiting and iterative mapping approach.
*Nucleic Acids Res* **41**(13), e129. (doi: 10.1093/nar/gkt371)
29 355
- Howe K et al. 2013 The zebrafish reference genome sequence and its relationship to the human
genome. *Nature* **496**, 498. (doi:<https://doi.org/10.1038/nature12111>)
- Jónsson H, Ginolhac A, Schubert M, Johnson PLF, Orlando L 2013 mapDamage2.0: fast approximate
Bayesian estimates of ancient DNA damage parameters. *Bioinformatics* **29**, 1682–1684. (doi:
10.1093/bioinformatics/btt193)
- Karimi K et al. 2018 Xenbase: a genomic, epigenomic and transcriptomic model organism database.
*Nucleic acids Res* **46**, 861–868. (doi: 10.1093/nar/gkx936)
- Katoh K, Misawa K, Kuma KI, Miyata T 2002 MAFFT: a novel method for rapid multiple
sequence alignment based on fast Fourier transform. *Nucleic Acid Res* **30**(14), 3059–3066.
(doi: 10.1093/nar/gkf436)
- Katoh K, Kuma KI, Toh H, Miyata T 2005 MAFFT version 5: improvement in accuracy of multiple
sequence alignment. *Nucleic Acid Res* **33**(2), 511–8. (doi: 10.1093/nar/gki198)
- Knuckey JDS, Ebert DA, Burgess GH 2011 *Etmopterus jounqi* n. sp., a new species of lanternshark
(Squaliformes: Etmopteridae) from Taiwan. *Aqua, International Journal of Ichthyology* **17**
(2), 61–72

Lassmann et al. 2010 SAMStat: monitoring biases in next generation sequencing data.
*Bioinformatics* **27**(1), 130-131. (doi: 10.1093/bioinformatics/btq614)
Leigh JW, Bryant D 2015 PopART: Full-feature software for haplotype network construction.
*Methods Ecol Evol* **6**(9), 1110–1116.
- Li H, Durbin R 2009 Fast and accurate short read alignment with Burrows-Wheeler transform.
*Bioinformatics* **25**, 1754–1760. (doi: 10.1093/bioinformatics/btp698)
Li H, Handsaker B, Wysoker A, Fennell T, Ruan J, Homer N, ..., Durbin R 2009 The sequence
alignment/map format and SAMtools. *Bioinformatics*, **25**, 2078–2079. (doi:
10.1093/bioinformatics/btp352)
- Lyra ML et al 2020 High-throughput DNA sequencing of museum specimens sheds light on the long-
missing species of the Bokermannohyla claresignata group (Anura: Hylidae: Cophomantini).
*Zool J Linn Soc* **190** (4), 1–21.
Martin M 2011 Cutadapt removes adapter sequences from high-throughput sequencing reads.
*EMBnet Journal* **17**(1), 10-12. (doi: 10.14806/ej.17.1.200)
- Morato T, Watson R, Pitcher TJ, Pauly D 2006 Fishing down the deep. *Fish and Fisheries*, **7**, 24–
34.
Murray BW, Wang JY, Yang SC, Stevens JD, Fisk A, Svavarsson J 2008 Mitochondrial cytochrome
b variation in sleeper sharks (Squaliformes: Somniosidae). *Marine Biology* **153**, 1015–1022.
(doi: 10.1007/s00227-007-0871-1)
- Naylo GJP, Caira JN, Jensen K, Rosana KAM, White WT Last PR 2012 A DNA sequence based
approach to the identification of shark and ray species and its implications for global
elasmobranch diversity and parasitology. *Bulletin of the American Museum of Natural History*,
**367**, 1-262.
- Naylor GJP, Ryburn JA, Ferigo O, Lopez A 2005 Phylogenetic relationships among the major
lineages of modern elasmobranchs. In: Hamlett, WC (ed), *Reproductive Biology and*
*Phylogeny of Chondrichthyes: Sharks, Batoids and Chimaeras*. Science Publishers, Enfield, pp.
1–25.
Paijmans JLA, Baleka S, Henneberger K, Taron U, Trinks A, Westbury M, Barlow A 2017
Sequencing single-stranded libraries on the Illumina NextSeq 500 platform.
[arXiv:1711.11004v1](https://arxiv.org/abs/1711.11004v1) [q-bio.OT]
- Pollerspöc J, Straube N 2020 Bibliography database of living/fossil sharks, rays and chimaeras
(Chondrichthyes: Elasmobranchii, Holocephali) -Papers of the year 2019, [www.shark-](http://www.shark-references.com)
[references.com](http://www.shark-references.com), WorldWide Web electronic publication, Version 01/2020;ISSN: 2195-6499

Rancilhac L, Bruy T, Scherz MD, Pereira EA, Preick M, Straube N,...Vences M 2020 Targeted-
405 enrichment DNA sequencing from historical type material enables a partial revision of the
Madagascar giant stream frogs (genus *Mantidactylus*). *J Nat Hist* **54**, 87-118.
(doi:[10.1080/00222933.2020.1748243](https://doi.org/10.1080/00222933.2020.1748243))
- Read TD et al. 2017 Draft sequencing and assembly of the genome of the world's largest fish, the
whale shark: *Rhincodon typus* Smith 1828. *BMC Genomics* **18**(1), 532. (doi: 10.1186/s12864-
410 017-3926-9)
Rogers AD, Gianni M 2010 The Implementation of UNGA Resolutions 61/105 and 64/72 in the
Management of Deep-Sea Fisheries on the High Seas. Report prepared for the Deep-Sea
Conservation Coalition. International Programme on the State of the Ocean, London, United,
Kingdom, 97pp.
Rohland N, Siedel H, Hofreiter M (2004). Nondestructive DNA extraction method for
mitochondrial DNA analyses of museum specimens. *Biotechniques* **36**(5), 814-821. (doi:
10.2144/04365ST05)
- Rohland N, Siedel H, Hofreiter M 2010 A rapid column-based ancient DNA extraction method for
increased sample throughput. *Mol Ecol Resour* **10**, 677–683. (doi: 10.1111/j.1755-
420 0998.2009.02824.x)
- Shirai S, Tachikawa H 1993 Taxonomic resolution of the *Etmopterus pusillus* species group
(Elasmobranchii, Etmopteridae), with description of *E. bigelowi*, n. sp. *Copeia* **2**, 483–495.
Straube N, Duhamel G, Gasco N, Kriwet J, Schliewen UK 2011a Description of a new deep-sea
lanternshark *Etmopterus viator* sp. nov. (Squaliformes: Etmopteridae) from the Southern
Hemisphere. In: Duhamel G, Welsford D (eds.) The Kerguelen Plateau, Marine Ecosystem
and Fisheries. Société Française d'Ichtyologie, Paris, pp. 135–148.
- Straube N, Kriwet J, Schliewen UK 2011b Cryptic diversity and species assignment of large
lanternsharks of the *Etmopterus spinax* clade from the Southern Hemisphere, (Squaliformes,
Etmopteridae). *Zool. Scr.* **40**, 61–75.
- Straube N, Iglésias SP, Sellos DY, Kriwet J, Schliewen UK 2010 Molecular phylogeny and node time
estimation of bioluminescent lanternsharks (Elasmobranchii: Etmopteridae). *Mol Phylogenet*
*Evol* **56**, 905–917. (doi.org/10.1016/j.ympev.2010.04.042)
- Straube N, Rochel E, Corrigan S, Li C, Naylor GJP 2015 On the occurrence of the Southern
Lanternshark, *Etmopterus granulosus*, off South Africa, with comments on the validity of *E.*
*compagnoi*. *Deep Sea Research Part II: Topical Studies in Oceanography* **115**, 11-17,
(doi.org/10.1016/j.dsr2.2014.04.004)

Straube N, White WT, Ho H-C, Rochel E, Corrigan S, Li C, Naylor GJP 2013 A DNA sequence-
based identification checklist for Taiwanese chondrichthyans. *Zootaxa*, **3752**, 256–278.
Straube N, Leslie RW, Clerkin PJ, Ebert DA, Rochel E, Corrigan S, Li C, Naylor GJP 2015 On the
occurrence of the southern Lanternshark, *Etmopterus granulosus*, off south Africa, with
440 comments on the validity of *E. compagnoi*. *Deep Sea Research Part II: Topical Studies in*
*Oceanography* **115**, 11–17.
Straube N, Lyra M, Paijmans JLA, Preick M, Basler N, Haddad CFB, Barlow A, Hofreiter M in
revision Successful application of ancient DNA extraction and library construction protocols to
museum wet collection specimens. *Mol Ecol Resour*
Veríssimo A, McDowell JR, Graves JE 2012 Genetic population structure and connectivity in a
commercially exploited and wide-ranging deepwater shark, the leafscale gulper (*Centrophorus*
*squamosus*). *Mar Freshwater Res*, **63** (6), 505–512. (doi: [10.1071/MF11237](https://doi.org/10.1071/MF11237))
Veríssimo A, McDowell JR, Graves JE 2011 Population structure of a deep-water squaloid shark, the
Portuguese dogfish (*Centroscymnus coelolepis*). *ICES J Mar Sci* **68**(3):555–563. (doi:
[10.1093/icesjms/fsr003](https://doi.org/10.1093/icesjms/fsr003))
Walter RP, Roy D, Hussey NE, Stelbrink B, Kovacs KM, Lydersen C, et al. 2017 Origins of the
Greenland shark (*Somniosus microcephalus*): Impacts of ice-olation and introgression. *Ecol*
*Evol* **7**, 8113–8125. (doi: [10.1002/ece3.3325](https://doi.org/10.1002/ece3.3325))
Wingett SW, Andrews S 2018 FastQ Screen: A tool for multi-genome mapping and quality control.
*F1000Research* **7**, 1338. (doi:10.12688/f1000research.15931.2)
Wyffels, J et al. 2013 Xenbase: a genomic, epigenomic and transcriptomic model organism database,
*Nature* **46**(1), 587–591. (doi: [10.1093/database/bar064](https://doi.org/10.1093/database/bar064))
Yano K, Stevens JD, Compagno LJV 2004 A review of the systematics of the sleeper shark genus
*Somniosus* with redescriptions of *Somniosus* (*Somniosus*) *antarcticus* and *Somniosus*
(*Rhinoscyrnus*) *longus* (Squaliformes: Somniosidae). *Ichthyol Res* **51**, 360–373.

Acknowledgements

We would like to express our sincere thanks to Ernst Mikschi and Anja Palandacic at the Museum of
Natural History in Vienna for the sampling opportunity of the *E. pusillus* syntype. Marianna Lyra,
Axel Barlow and Johanna Paijmans are cordially acknowledged for their recommendations regarding
data analysis.

Funding

This work was funded by the German Research Foundation (DFG; grants STR-1429 1/1 to NS and HO-3492 7/1 to MH within the DFG SPP 1991 “Taxon-Omics”). This project was further supported by the National Science Foundation (NSF), grant “Jaws and Backbone: Chondrichthyan Phylogeny and a Spine for the Vertebrate Tree of Life”; DEB-01132229 to GJPN.

Data accessibility

in preparation

Competing interests

The authors declare no competing interests.

Author contributions

NS and MH designed the study. NS and MP conducted laboratory steps. NS and MH analysed the data. GJP provided NADH2 sequences for the network analysis. NS wrote and drafted the manuscript with contributions from all authors.

Table & Figure Captions:

Table 1: Overview of samples used for the NADH2 haplotype network reconstruction.

Table 2: Summary statistics for the reconstructed mitochondrial genome sequence of the *Etmopterus pusillus* syntype NMW- 78526-15.

Figure 1: Fastqscreen analysis for the 116 year old *E. pusillus* syntype archival DNA. From a subsample of 99,859 sequencing reads, 6,971 reads map exclusively to the *E. spinax* transcriptome derived from the study by Delroisse et al. (2019) confirming the presence of target DNA.

Figure 2: Single read length distribution of 8,970,773 trimmed reads derived from the *E. pusillus* syntype specimen. The majority of reads is below 50 bp.

Figure 3: DNA damage patterns of the 116 year old *E. pusillus* syntype archival DNA. Sequencing reads show increased C to T (red) and G to A substitutions (blue) at the 3' ends. The 5' prime ends also show a slight increase in substitution rates. Grey: other substitutions. A: Damage pattern

estimates from total reads. B: Damage pattern estimates from the mitochondrial reads used for the
reconstruction of the mitochondrial genome.

Figure 4: Per base coverage plot of the *Etmopterus pusillus* syntype NMW- 78526-15 mitochondrial
genome. The blue line shows the median.

Figure 5: Annotated circular mitochondrial genome sequence of the *Etmopterus pusillus* syntype
NMW- 78526-15. Annotation reference: Genbank accession KU892588.

Figure 6: Haplotype network of NADH2 sequences derived from *E. pusillus*, *E. jounqi* and the *E.*
*pusillus* syntype specimen. Numbers in brackets indicate mutational steps.

Table 1: Overview of samples used for the NADH2 haplotype network reconstruction.

Number	Genus	Species	Locality	sample number	Genbank Accession
Etmopterus	pusillus	East China Sea	KU892588	in preparation
Etmopterus	joungi	Changbin, Taitung, Taiwan	GN10144	in preparation
Etmopterus	joungi	Changbin, Taitung, Taiwan	GN10145	in preparation
Etmopterus	joungi	Changbin, Taitung, Taiwan	GN10148	in preparation
Etmopterus	pusillus	Azores, Atlantic Ocean, Portugal	GN6543	in preparation
Etmopterus	pusillus	Azores, Atlantic Ocean, Portugal	GN6548	in preparation
Etmopterus	pusillus	Azores, Atlantic Ocean, Portugal	GN6550	in preparation
Etmopterus	pusillus	Azores, Atlantic Ocean, Portugal	GN6552	in preparation
Etmopterus	pusillus	Azores, Atlantic Ocean, Portugal	GN6555	in preparation
Etmopterus	pusillus	Azores, Atlantic Ocean, Portugal	GN6563	in preparation
Etmopterus	pusillus	Azores, Atlantic Ocean, Portugal	GN6578	in preparation
Etmopterus	pusillus	Eastern North Atlantic, Portugal	GN6603	in preparation
Etmopterus	pusillus	Eastern North Atlantic, Portugal	GN6620	in preparation
Etmopterus	pusillus	Suruga Bay, North Pacific, Japan	GN7426	in preparation
Etmopterus	pusillus	E of Tweed Heads, Border Bank Site, NSW, Australia	GN11335	in preparation
Etmopterus	pusillus	E of Tweed Heads, Border Bank Site, NSW, Australia	GN11334	in preparation
Etmopterus	pusillus	E of Tweed Heads, Border Bank Site, NSW, Australia	GN11331	in preparation
Etmopterus	pusillus	E of Tweed Heads, Border Bank Site, NSW, Australia	GN11330	in preparation
Etmopterus	pusillus	E of Tweed Heads, Border Bank Site, NSW, Australia	GN11329	in preparation
Etmopterus	pusillus	Eastern North Atlantic, Portugal	GN3772	in preparation
Etmopterus	pusillus	Eastern North Atlantic, Portugal	GN3771	in preparation
Etmopterus	pusillus	Eastern North Atlantic, Portugal	GN3770	in preparation
Etmopterus	pusillus	Eastern North Atlantic, Portugal	GN3769	in preparation
Etmopterus	pusillus	Eastern North Atlantic, Portugal	GN3767	in preparation
Etmopterus	pusillus	Eastern North Atlantic, Portugal	GN3766	in preparation
Etmopterus	pusillus	Eastern North Atlantic, Portugal	GN3765	in preparation
Etmopterus	pusillus	Tasman Sea, NSW, Australia	GN2614	in preparation
Etmopterus	pusillus	Morocco, East Atlantic	GN12128	in preparation
Etmopterus	pusillus	Eastern North Atlantic, Portugal	GN6624	in preparation
Etmopterus	pusillus	Tasman Sea, NSW, Australia	GN2613	in preparation
Etmopterus	pusillus	Tasman Sea, NSW, Australia	GN4951	in preparation
Etmopterus	pusillus	Tasman Sea, NSW, Australia	GN11328	in preparation
Etmopterus	pusillus	Suruga Bay, North Pacific, Japan	GN7436	in preparation
Etmopterus	pusillus	Suruga Bay, North Pacific, Japan	GN7435	in preparation
Etmopterus	pusillus	Tasman Sea, Norfolk, Australia	GN7396	in preparation

Table 1: Summary statistics for the reconstructed mitochondrial genome sequence of the *Etmopterus pusillus* syntype NMW- 78526-15.

Length (bp)	GC content (%)	A (%)	C (%)	G (%)	T (%)	N (%)	Average assembly quality	Average per base coverage
16741	37.2	31.1	22.8	14.3	31.5	0.4	73	12.12

Figure 1: Fastqscreen analysis for the 116 year old *E. pusillus* syntype archival DNA. From a subsample of 99,859 sequencing reads, 6,971 reads map exclusively to the *E. spinax* transcriptome derived from the study by Delroisse et al. (2019) confirming the presence of target DNA.

Figure 2: Single read length distribution of 8,970,773 trimmed reads derived from the *E. pusillus* syntype specimen. The majority of reads is below 50 bp.

Figure 3: DNA Damage patterns of the 116 year old *E. pusillus* syntype archival DNA. Sequencing reads show increased C to T (red) and G to A substitutions (blue) at the 3' ends. The 5' prime ends also show a slight increase in substitution rates. Grey: other substitutions.

A: Damage pattern estimates from total reads.

B: Damage pattern estimates from the mitochondrial reads used for the reconstruction of the mitochondrial genome.

A total reads

B mapped mitochondrial reads

Figure 4: Per base coverage plot of the *Etmopterus pusillus* syntype NMW- 78526-15 mitochondrial genome. The blue line shows the median.

1 Figure 5: Annotated circular mitochondrial genome sequence of the
 2 *Etmopterus pusillus* syntype NMW- 78526-15.
 3

4 Annotation reference: Genbank accession KU892588.

Figure 6: Haplotype network of NADH2 sequences derived from *E. pusillus*, *E. joungi* and the *E. pusillus* syntype specimen. Numbers in brackets indicate mutational steps.

Appendix B

Bergen, xx.06.2021

To
Lianne Parkhouse
Editorial Coordinator
Royal Society Open Science

Dear Dr. Parkhouse,

Thank you very much for considering our manuscript for publication in RSOS. Please find a point-to-point response to the reviewers below. After implementing several changes to improve the manuscript, we hope that we meet all the reviewers' concerns and would be happy if you can accept the manuscript for publication in RSOS.

With best wishes,

on behalf of all authors,

Nicolas Straube

Reviewer comments to Author:

Reviewer: 1

Comments to the Author(s)

This paper is a valuable contribution to the field of taxonomy and highlights the great potential in using old specimens in museum wet-collections in clarifying some pending taxonomic questions. The paper is interesting to read, is scientifically sound and draws a simple but a valuable conclusion to Etmopterus taxonomy, but I think a change in the framing of the work would make the paper more appealing to a broader audience. Specifically, rather than asking one taxonomic question on one specific genus (and how the authors generated the data to answer it), the paper can be framed as presenting a simple methodological approach to answer pending taxonomic questions using valuable but underused museum specimens from old wet collections. Major changes would be needed mostly in the Introduction.

R: Thank you for pointing this out. We re-structured and partially re-wrote the introduction to emphasize the wider scope of pending taxonomic questions (lines 36-50). Details on the method, however, are available in Straube & Lyra et al. (2021).

Straube, N., Lyra, M.L., Paijmans, J.L.A., Preick, M., Basler, N., Penner, J., Rödel, M.-O., Westbury, M.V., Haddad, C.F.B., Barlow, A. and Hofreiter, M. (2021), Successful application of ancient DNA extraction and library construction protocols to museum wet collection specimens. Molecular Ecology Resources. Accepted Author Manuscript. <https://doi.org/10.1111/1755-0998.13433>

Minor changes/edits and comments were made in the pdf file of the submitted manuscript, in attachment.

R: Thank you, all minor changes and comments have been addressed, please see the tracked changes version of the manuscript.

Reviewer: 2

Comments to the Author(s)

This paper reports the NGS sequencing of a syntype sample of *E. pusillus*, the extraction of mitochondrial sequences from the library of recovered sequences, the Sanger sequencing of NADH2 from a number of recently collected samples and evaluation of the NADH2 haplotype network for evidence to support the recently described species, *E. jounqi*.

The data collection is well done, although I have one concern (see below). The analysis of the 116 wet-preserved samples is impressive and the cost of 37 euros should allow other syntype samples to be sequenced. In fact, this low cost makes me wonder why only 1 syntype sample was included in this study.

R: Thank you for this positive evaluation. The sample was initially part of a larger experiment comprising several samples from very different taxa covering a diversity of vertebrate wet-collection specimens (Straube & Lyra et al. 2021). We were therefore not able to include more samples from the syntype series. Besides, the syntype series stems from the same location and type material in general should be minimally sampled to spare morphological characters. The cost of 37,- Euros refers to the costs for sequencing at the University of Potsdam and does not include the costs for DNA extraction and library preparation in a dedicated historical laboratory. To avoid misinterpretation of these costs, we deleted this statement from the text as also reviewer 1 was confused by it.

Straube, N., Lyra, M.L., Paijmans, J.L.A., Preick, M., Basler, N., Penner, J., Rödel, M.-O., Westbury, M.V., Haddad, C.F.B., Barlow, A. and Hofreiter, M. (2021), Successful application of ancient DNA extraction and library construction protocols to museum wet collection specimens. Molecular Ecology Resources. Accepted Author Manuscript. <https://doi.org/10.1111/1755-0998.13433>

My main concerns with the paper arise from the sample collection design to meet the stated objectives of the paper, the provenance and species ID of the samples included for the analysis, and relatively basic interpretation of the phylogeographic results. I will outline these below.

1) The title of the paper reads, "Sequencing of the mitochondrial genome of a 116 year old wet collection syntype confirms an almost global distribution of the smooth lantern shark". I have problem with this title as I don't see why the sequencing of a single sample would confirm the global distribution of a species? This point is never fully discussed in the paper. Yes, the syntypes NADH2 haplotype is within the diversity seen in samples collected more recently at various locations around the globe, but how does this syntype sample, in itself, confirm the Global distribution?

R: Thank you for this comment. We changed the title to 'Mitochondrial DNA sequencing of a wet-collection syntype demonstrates the importance of type material as genetic resource for Lantern Shark taxonomy (Chondrichthyes: Etmopteridae)'.

2) A number of syntypes for *E. pusillus* exist (8 samples, line 94). Why was this particular samples chosen for analysis?

R: Please note that sampling options of type material are often limited to spare morphological characters and collection curators decide how many and which samples can be sampled. The sample used herein showed an external latero-dorsal abrasion, which was an ideal spot for sampling. The sampling was performed by the staff of the Vienna Museum.

Why not all 8 if you want to confirm the global distribution of the species?

R: All syntype specimens of *E. frontimaculatus* stem from the same locality in the North Pacific (off Yokohama, Japan). We added this information in the text (lines 101-102) and Table 1. We did include NADH2 sequences in our network analyses of *E. pusillus* from close to the type locality of *E. pusillus*, which is the North Atlantic (Madeira, Portugal).

I presume this is because the sample was collected off Japan (closest geographically to the current *E. joungi* samples), but this point is not address in the text. As a reader I have no context with which to judge the importance of this sample. For example, how does this sample relate to the study of Knuckey et al 2011, which describes *E. joungi*?

R: We included detailed information on the taxonomic history of *E. pusillus*, *frontimaculatus* and *E. joungi* in the introduction (lines 64 – 84): The Smooth Lanternshark was originally described from Madeira in the Northeast Atlantic Ocean. After its description in 1839, morphologically similar specimens were reported from numerous geographically distant locations far from the type locality suggesting the species was widespread. Shirai & Tachikawa (1993) were the first to explore the taxonomic status of the morphologically similar specimens. These authors described a new species, *E. bigelowi* and commented on the distribution range of *E. pusillus* as occurring almost globally by declaring the Northwest Pacific species *E. frontimaculatus* Pietschmann 1907 as a junior synonym to *E. pusillus*. In 2011, however, Knuckey et al. described a morphologically similar species from the Northwest Pacific (Taiwan), the Shortfin Smooth Lanternshark *E. joungi*. The species is described to occur localized off Taiwan. Later studies based on the analyses of mitochondrial NADH dehydrogenase subunit 2 gene sequences showed that mitochondrial haplotypes of *E. pusillus* from both the Pacific and Atlantic Ocean mix along with specimens identified as *E. joungi* from Taiwan (Straube et al. 2013). The authors suggested that the species status of *E. joungi* be reviewed to clarify whether *E. joungi* represented a sub-population of *E. pusillus* or indeed a distinct species. Here, we applied an ancient DNA approach to sequence archival mitochondrial DNA of one of the syntypes of *E. pusillus* collected in 1903 in the Northwest Pacific off Japan, which represents a specimen of the former type series of *E. frontimaculatus*.

Does it clearly show it belongs to the description of *E. pusillus* and not *E. joungi*?

R: The *E. frontimaculatus* was synonymized with *E. pusillus* based on their morphology in Tachikawa et al. 1993. Not before 2011 Knuckey et al. described *E. joungi* morphologically separating it from *E. pusillus*. They did not inspect the North Pacific syntypes of *E. pusillus* for their analysis, however. Nevertheless, at this point, the former *E. frontimaculatus* types have to be regarded as *E. pusillus*. Our results clearly support the synonymy of *E. frontimaculatus* and *E. pusillus*.

It was collected about 100 years before Knuckey et al (2011) which analyzed samples collected north east of Taiwan (but actually from fish markets in Taiwan). Japan also lies north east of Taiwan.

R: Our network analysis includes haplotypes from a diversity of localities comprising the North Atlantic as well as the Southwest and Northwest Pacific. The network shows that Southwest Pacific haplotypes are spread out within the network in close proximity to North Pacific samples as well as *E. joungi*. The exact sampling locality of *E. joungi* is unknown.

Do the records show exactly where the syntype was originally collected?

R: The sample was collected off Yokohama (Japan) in the Northwest Pacific. We added this information in the main document lines 101-102 and in Table 1. Figure 6 as well as the introduction further provide information on the syntype.

Is there a possibility that the samples come from the same geographic population (East China Sea) of sharks?

R: As aforementioned, the sample of the *E. pusillus* syntype was collected east off Japan (Yokohama), which is not part of the East China Sea.

3) In general, please provide more context for the samples used for the study. I understand that these samples are hard to come by and won't represent the entire global range, but more effort should be made to show the distribution and sampling details of the samples.

R: Thank you for pointing this out. We added a sampling map as new figure 1 to the manuscript.

Table 1 is an unorganised list of the samples. This should be organized by species and then by geographic location.

R: Done.

Samples new to this study should also be indicated. A number of samples/sequences were from Straube et al 2013. These should be indicated.

R: Done.

Where possible the GPS (or relative) locations should be indicated, along with collection dates. I.e., how do the Tweed Heads, NSW samples differ from the Tasman Sea, NSW samples?

R: We included all available information. Unfortunately, no more detailed information on localities is available. We hope that Figure 1 can serve as an overview of sampling locations.

Sample 1 is from the East China Sea. The Taiwan samples would also be from the East China Sea.

R: Not necessarily, as they may well be sampled in the Phillipine Sea. The samples were collected on the same fish market as the *E. jounji* type specimens. The exact catching location is unknown. We added this information in Table 1.

Do these samples represent the different morphotypes from the same environment (more on the importance of this sample below)?

R: We are not sure what the reviewer is referring to. Even if we would doubt the integrity of Chen et al's (2016) species identification, our results argue strongly in favor of a junior synonymy of *E. jounji* with *E. pusillus*.

4) Following this point, and as species ID is really important for correct interpretation of phylogeographic patterns, please indicate how you determined the species ID of the samples in the North East Pacific. I.e., Was species determination conducted in light of the recent description of *E. jounji*?

R: We do not have samples from the Northeast Pacific. If the reviewer is referring to the Northwest Pacific, the samples from Suruga Bay were collected and identified by Sho Tanaka and NS in 2007. The identification was not reviewed as no voucher specimen was preserved and *Etmopterus jounqi* was hitherto only known from Northeast of Taiwan (quote from Knuckey et al. (2011): Distribution: Holotype and paratypes were collected along the upper continental slope by bottom trawler off north-eastern Taiwan at a depth greater than 300 m.). The additional *E. jounqi* samples were collected and identified by D. Ebert (Co-Author of Knuckey et al. 2011) and NS mainly based on their sampling location in 2012.

For example, Sample 1 is listed as *E. pusillus*, collected in the East China Sea and listed as sample number KU892588.

R: As the sample was collected and sequenced in 2016, 6 years after the publication of *E. jounqi*, we assume that Chen et al. (2016) were aware of *E. jounqi* and applied corresponding identification measures to their sample.

In fact, the sample number is the Genbank accession number (noted as “in preparation”) for the whole mtDNA genome of a sample reported in Chen et al (2016).

R: Thank you for pointing this out, we corrected Table 1.

Chen et al described the sample provenance as “One specimen of *E. pusillus* was captured from continental shelf in East China Sea and landed on a pier in Wenling, Zhejiang, China. It was preserved in the museum of marine biology in Wenzhou Medical University with voucher WL2012051264.” No dates are given for the sample collection and consider that Wenling is much closer to Taiwan than Japan. Note, travelling west of Wenling on the East China Sea takes you to just northwest of Taiwan, around the same location where the samples used to describe *E. jounqi* were reported to be collected (Knuckley et al 2011).

R: As aforementioned, the *E. jounqi* specimens in Knuckey et al. (2011) were collected north-eastern off Taiwan and may stem from the Phillipine Sea.

Note also, if this sample was collected prior to 2011, it would have, by default been listed as *E. pusillus*. Please indicate how you determined it was actually *E. pusillus* and not *E. jounqi*?

R: As Chen et al. published in 2016, we assume that Chen et al. were identifying the specimen taking *E. jounqi* into account. If Chen et al. (2016) mis-identified the specimen, our results would not be influenced by it.

5) The haplotype network does not indicate which samples correspond to which haplotypes. Although it is relatively standard not to include labels in a network, I suggest that you add the haplotype codes to Table 1 so that a comparison be made.

R: We added the information in Figure 6.

Because I don't know which samples share haplotypes the following is part conjecture on my part. I noted that the syntype matches one of the North East Pacific samples. As the East China Sea haplotype (discussed in point 4) was used as the reference for the syntype reconstruction, can you please comment on whether these two samples differ at the NADH2 locus used for the phylogeographic comparison. If they differ great, but please note this in the results. However, if they match, I suspect there may be a bias associated with the reference in determining which mutations

are real and which are artifacts. Six unique NADH2 haplotypes were found among seven recent North East Pacific samples (both species). Clearly more haplotypes are yet to be sampled in this geographic region. The odds that the syntype would match one already sampled are low. The odds that it would match the reference mtDNA are even lower. So, if they match, I am suspicious of a methodological bias in the determination of the syntype sequence.

R: Please note that the syntype sequence is not matching one but two haplotypes. One sample is a specimen initially recorded as *E. joungi* (GN10144), while the other is an *E. pusillus* specimen from Suruga Bay, Japan (GN7435). The specimen from Chen et al. (2016; KU892588) clusters with *E. joungi* (GN10145, GN10148) as well as *E. pusillus* samples from the Southwest and Northwest Pacific. There is no match between the reference haplotype and the syntype haplotype. However, the iterative mapping approach in MITObim also uses the reference KU892588 only initially in the first iteration to map a first set of reads. These mapping hits are then used in the next iteration as reference to facilitate further read mapping (Hahn et al. 2013), i.e. the detected similarities/differences are real.

Hahn C, Bachmann L, Chevreux B 2013 Reconstructing mitochondrial genomes directly from genomic next-generation sequencing reads—a baiting and iterative mapping approach. *Nucleic Acids Res* 41(13), e129. (doi: 10.1093/nar/gkt371)

Note, if these samples match, then they also match one of the *E. joungi* samples. This is the only case of haplotype sharing in the North East Pacific. Speculation: if the East China Sea sample is actually another sample of *E. joungi*, and there are doubts to the accuracy of the syntype haplotype, then you would need to reassess your conclusions.

R: Even if the haplotypes are not matching (see response above), we disagree with the reviewer here. Our network is color coded by geography and clearly shows close relatedness of haplotypes from the North and Southwest Pacific, partially separated by only two mutational steps. A distinct species should show some clearer delimitation from its closest relative based on the work which has been conducted in sharks using NADH2 so far (e.g. Naylor et al. 2012; Straube et al. 2013). Adding the hitherto phylogenetically closest known sister species of *E. pusillus*, *E. bigelowi*, to the network analysis shows 71 mutational steps difference to *Etmopterus pusillus* demonstrating species level delimitation.

Naylor GJP, Caira JN, Jensen K, Rosana KAM, White WT Last PR 2012 A DNA sequence based approach to the identification of shark and ray species and its implications for global elasmobranch diversity and parasitology. *Bulletin of the American Museum of Natural History*, 367, 1-262.

Straube N, White WT, Ho H-C, Rochel E, Corrigan S, Li C, Naylor GJP 2013 A DNA sequence-based identification checklist for Taiwanese chondrichthyans. *Zootaxa*, 3752, 256–278.

6) I find the discussion too brief and the treatment of the evidence for or against the validity of *E. joungi* mostly lacking. (Not having a good understanding of the provenance of the samples does not help my understand - covered above). I agree, the data as presented clearly does not provide support for *E. joungi*. The reciprocal monophyly noted in many other groups does not exist here. However, the discussion should focus on the expectations of lineage sorting and not rely on a reference to a paper that this gene works well to sort out sharks in other genera.

R: We added a section discussing that haplotypes are mixing and do not cluster on species level, as would be expected for two distinct species (now lines 286-290).

We are citing the adequate NADH2 papers, as this locus is not only working well in species delimitation within the same genus *Etmopterus* (e.g. Straube et al. 2013) but across all *Chondrichthyes* (e.g. Naylor et al. 2012). We consider the locus therefore well-established.

Naylor GJP, Caira JN, Jensen K, Rosana KAM, White WT Last PR 2012 A DNA sequence based approach to the identification of shark and ray species and its implications for global elasmobranch diversity and parasitology. Bulletin of the American Museum of Natural History, 367, 1-262.

Straube N, White WT, Ho H-C, Rochel E, Corrigan S, Li C, Naylor GJP 2013 A DNA sequence-based identification checklist for Taiwanese chondrichthyans. Zootaxa, 3752, 256–278.

Based on the polyphyletic stage of the lineage sorting among two taxa examined, this haplotype network would suggest either an evolutionary recent speciation event or a single species. What is sum of evidence for either of these two hypotheses.

R: Our results do not argue in a recent speciation event as haplotypes separated by only few mutational steps from geographically distant locations are in close proximity. Note also that Southwest Pacific haplotypes cluster with North Atlantic haplotypes.

The two branches of the network which do not include North Atlantic haplotypes are separated by a maximum of 4 and 7 mutational steps from the branch including North Atlantic haplotypes, which in itself contains mutational steps of more than 8, even though samples come from the same Northeast Atlantic area. Please note further that the now included *E. bigelowi* is the sister species of *E. pusillus* and shows a difference of 71 mutational steps, which exemplifies NADH2 differences on species level in the *E. pusillus* species group.

Key here are the samples from the North East Pacific. Are both species types currently found East China Sea?

R: Our results argue in favor of a single species.

Is this a regional morphotype of the same species or do the morphotypes overlap?

R: Our results argue in favor of a single species distributed in the area, i.e., *E. pusillus*. We did not analyse morphological variation in this study. Knuckey et al. (2011) report on geographic morphological variation within *E. pusillus*. Ontogenetic and/or sexual differences were not analysed so far., however, we conclude that the *E. jounqi* morphology represents intraspecific variation within *E. pusillus*.

Is there any evidence to suggest a lack of gene flow with the other locations?

R: Our results argue in favor of geneflow between distant locations as discussed in lines 270 – 276.

How do these findings relate to the samples used in the original description (Knuckey et al 2011)? Does Knuckey's paper also include other samples from the North East Pacific for its morphological comparison? Does it suggest the likely range distribution of the morphotype associated with *E. jounqi*? The entire East China Sea, or just around Taiwan? How does the sequence of the old (pre *E. jounqi* description) syntype relate to the interpretations? Does its original description fit the more recent description of *E. jounqi*, or is it more like *E. pusillus* samples used in the Knuckey paper? How does this all relate to the original description of *E. frontimaculatus* in this area? Lastly, what other studies could be done to resolve this question?

R: We expanded our discussion in lines 287-290 to provide more details on Knuckey et al's (2011) work.

Minor points

1) A comparison with Straube et al (2013), shows three *E. jounqi* samples were analyzed in this earlier paper. The current study also uses three samples, two are the same but GN10140 is not used and a new sample, GN10148 is used instead. Given the paucity of data on this species, why would you exclude this previous sample? Or is this typo in Straube et al, Fig 2, or in the present table 1?

R: Thank you for pointing this out. We added GN10140 to the dataset, please see Figure 6 and Table 1.

2) Figure 2 shows the distribution of NGS fragment sizes. The range is from 22 to 72+ bp and the average is reported as 30.5. This average is described in the results after the reporting of the final number of trimmed and filtered reads 8,970,773 (line 212). This is a bit confusing, as the trimmed sequences should not contain any reads below 30 bp (line 113/4). As Figure 2 appears to show the prefilter/trimmed reads, it should be reported prior to the reporting of the final number of sequences used to assemble the mtDNA genome. Also, please explain what figure 2 is actually showing. Are these prefiltered or postfiltered reads?

R: The first mentioning of 30.5 bp was referring to the insert size of the library. We inserted the correct average read length. We excluded reads <25 basepairs, which is now pointed out in lines 130. The figure caption is expanded for clarification now reading 'Figure 2: Single read length distribution of trimmed and quality filtered reads. Reads below a length of 25 basepairs were excluded. Most reads are below a length of 50 bp.'

3) Lines 296-301 describe the haplotype sharing observed among Pacific and Antarctic sleeper sharks. Is Edwards et al (2019) the correct reference? This paper is a review of the Greenland Shark. Is Murray et al (2008) the better reference?

R: Done.

4) Lines 58-59 state the *E. bigelowi* is a more distant relative to *E. pusillus* than *E. jounqi*. What is the reference/evidence for this statement?

R: Done. We added corresponding references.

Straube N, White WT, Ho H-C, Rochel E, Corrigan S, Li C, Naylor GJP 2013 A DNA sequence-based identification checklist for Taiwanese chondrichthyans. *Zootaxa*, 3752, 256–278.

5) No samples of *E. bigelowi* are included in this study. As another member of this "species group" these samples would likely help provide context for the interpretation of the *E. jounqi* samples. If these are available I would suggest adding them to your haplotype network.

R: Done. We added a NADH2 sequence of *E. bigelowi* to the network construction (Table 1, Figure 6).

6) Please double check your reference list. A quick review found a number of typos/errors. i.e., line 390, inconsistent use of short and long forms for journals, Straube et al 2015 is listed twice.

R: Done. RSOS initially does not request any type of format. We edited the references in Vancouver style, as indicated by RSOS after the revision.

Appendix C

Mitochondrial DNA sequencing of a wet-collection syntype demonstrates the importance of type material as genetic resource for Lantern Shark taxonomy

(Chondrichthyes: Etmopteridae)

RESPONSE TO REVIEWERS

-please note that all line information refers to the track changes version of the manuscript -

Reviewer: 1

Comments to the Author(s)

Dear Authors,

I have read the revised version of the MS and I believe the changes incorporated in the text following the two reviewer's suggestions have improved the quality of the results and of the MS overall.

R: thank you for the positive evaluation.

I have only two minor comments to make:

1. In line with the broader scope of the MS now, i.e. how generating reference sequence data from old, wet-collection specimens from Museum collections is important and can be made following your approach, I believe you can further reinforce the relevance of the paper by highlighting that you successfully generated key reference genetic sequence data for type specimens -which are key in solving some pending taxonomic questions, regarding for example taxonomicity validity of previous synonymy.

R: thank you for pointing this out. We added a sentence in lines 271-272 emphasizing the broader scope: "Our subsequent approach allowed for generating key reference data for answering pending taxonomic questions."

Indeed, you could have reinforced the fact that the data supported the synonymy of *E. frontimaculatus* as *E. pusillus* proposed by Shirai & Tachikawa (1993) based on morphological data.

R: We added this information in lines 299-301: "Our DNA sequence-based results support the synonymy of E. frontimaculatus with E. pusillus, first suggested by Shirai & Tachikawa (17) based on morphological evidence."

2. The symbols and colors of the new Figure 1 (map) could be changed to improve readability. For instance, you could use different symbols per species (e.g. circles, squares, diamonds). As is, you have 4 different colors to designate the same species, although the colors denote different sampling locations (not indicated in the legend but implied from the map). I would also include the reference of "*Etmopterus pusillus* syntype" to the top of the remainder *E. pusillus* references, or simply ordering the color/symbol legend alphabetically. If possible, the same symbols and colors should be consistent between the new Figures 1 and 6.

R: done.

Reviewer: 2

Comments to the Author(s)

I find the revised manuscript to be significant improvement from the first submission and am generally happy with the authors responses to my initial concerns that mostly arose from the lack of clarity of results. I agree with the revised title and feel the tables and figures (especially Fig 6) have been much improved. I only have a have few minor points for consideration.

R: Thank you for the positive feedback on our revised version.

Minor points.

1) Word usage, mostly in the revised text, could be improved through another round of edits. The first sentence of the abstract is awkwardly written. See also: page 3, last sentence of second paragraph; page 3, first sentence of third paragraph; page 11, consider replace the word “However” which starts the first paragraph with “Despite the observed DNA damage”.

R: done.

2) The replacement of “Northeast” for “North East”, etc. was only partially completed in the revised manuscript.

R: checked and improved.

3) I am not sure your removal of authors names in the text and replacement with bracketed references is correct. E.g., on page 4 – “In 2011, however, (18) described a morphological similar species ...”, I think this should read “In 2011, however, Knuckey et al. (18) described a morphological similar species ...” A scan of recent articles in this journal shows the later usage. Which reads a lot nicer - :-).

R: checked and improved.

4) page 12. I don't find the end of the second last paragraph clear. The logic sounds like your conclusion that the morphological differences noted for E jounqi must be part of the wider intraspecific variation is what leads to your recommendation that E jounqi is a junior synonym. This recommendation is actually based solely on the mtDNA results, which leads to a reinterpretation of Knuckley's results. Consider revising. E.g., The expectation of monophyly is not met for E. jounqi. ... Our results suggest that the detected morphological differences noted between E jounqi and the comparator E pusillus samples (18) represent the range of intraspecific variation in this widespread species and that E jounqi is a junior synonym of E pusillus. Or something to this effect. Perhaps I am overthinking this, but the way it was written didn't sound quite right to me.

R: we re-wrote the sentence for clarification. It now reads “In their description of E. jounqi, Knuckey et al. (18) report on intraspecific geographic morphological variation within their comparative material of E. pusillus analysed. The detected morphological differences between E. pusillus and E. jounqi in Knuckey et al. (18) may therefore represent further intraspecific variation. We therefore suggest that E. jounqi is a junior synonym of E. pusillus. Our DNA sequence-based results support the synonymy of E. frontimaculatus with E. pusillus, first suggested by Shirai & Tachikawa (17) based on morphological evidence.” (lines 295-301)